# SMN deficiency perturbs monoamine neurotransmitter metabolism in spinal muscular atrophy

Valeria Valsecchi[1,9], Francesco Errico [2,3,9], Valentina Bassareo[4,9], Carmen Marino [5,9], Tommaso Nuzzo[3,6], Paola Brancaccio[1], Giusy Laudati[1], Antonella Casamassa[7], Manuela Grimaldi[5], Adele D'Amico[8], Manolo Carta[4], Enrico Bertini[8], Giuseppe Pignataro[1], Anna Maria D'Ursi[5] & Alessandro Usiello [3,6✉]

Beyond motor neuron degeneration, homozygous mutations in the *survival motor neuron 1* (*SMN1*) gene cause multiorgan and metabolic defects in patients with spinal muscular atrophy (SMA). However, the precise biochemical features of these alterations and the age of onset in the brain and peripheral organs remain unclear. Using untargeted NMR-based metabolomics in SMA mice, we identify cerebral and hepatic abnormalities related to energy homeostasis pathways and amino acid metabolism, emerging already at postnatal day 3 (P3) in the liver. Through HPLC, we find that SMN deficiency induces a drop in cerebral nor-epinephrine levels in overt symptomatic SMA mice at P11, affecting the mRNA and protein expression of key genes regulating monoamine metabolism, including aromatic L-amino acid decarboxylase (AADC), dopamine beta-hydroxylase (DβH) and monoamine oxidase A (MAO-A). In support of the translational value of our preclinical observations, we also discovered that SMN upregulation increases cerebrospinal fluid norepinephrine concentration in Nusinersen-treated SMA1 patients. Our findings highlight a previously unrecognized harmful influence of low SMN levels on the expression of critical enzymes involved in monoamine metabolism, suggesting that SMN-inducing therapies may modulate catechola-mine neurotransmission. These results may also be relevant for setting therapeutic approaches to counteract peripheral metabolic defects in SMA.

[1] Division of Pharmacology, Department of Neuroscience, Reproductive and Dentistry Sciences, School of Medicine, University of Naples "Federico II", 80131 Naples, Italy. [2] Department of Agricultural Sciences, University of Naples "Federico II", 80055 Portici, Italy. [3] Laboratory of Translational Neuroscience, Ceinge Biotecnologie Avanzate, 80145 Naples, Italy. [4] Department of Biomedical Sciences, University of Cagliari, 09042 Monserrato, Italy. [5] Department of Pharmacy, University of Salerno, 84084 Fisciano, Salerno, Italy. [6] Department of Environmental, Biological and Pharmaceutical Science and Technologies, Università degli Studi della Campania "Luigi Vanvitelli", 81100 Caserta, Italy. [7] IRCCS Synlab SDN, 80143 Naples, Italy. [8] Unit of Neuromuscular and Neurodegenerative Disorders, Bambino Gesù Children's Hospital IRCCS, 00163 Rome, Italy. [9] These authors contributed equally: Valeria Valsecchi, Francesco Errico, Valentina Bassareo, Carmen Marino. ✉email: usiello@ceinge.unina.it

Spinal muscular atrophy (SMA) is an autosomal recessive neuromuscular disease characterized by progressive muscle weakness and paralysis due to selective degeneration of anterior horn cells in the spinal cord and motor nuclei in the brainstem, leading to premature infant death in the absence of treatment[1]. SMA is caused by homozygous deletions or mutations in the *survival motor neuron 1* (*SMN1*) gene, which results in the deficiency of the ubiquitous SMN protein[2]. A second paralogue gene, *SMN2*, present in variable copy numbers, can only partially compensate for *SMN1* loss due to a nucleotide change that affects splicing and decreases the generation of the full-length protein[3].

SMN is critical in orchestrating spliceosomal small nuclear ribonucleoprotein (snRNP) assembly for major (U2-dependent) and minor spliceosomes[1–4]. In addition, SMN plays a prominent influence in regulating other cellular processes, including axonal transport and local translation at the synapse, cytoskeletal actin dynamics, synaptic vesicle release, endocytosis, autophagy, and mitochondrial bioenergetics, thus supporting a general housekeeping function for this protein[1,5–8].

SMA is classified into four main phenotypes (SMA1–4) according to the age of onset, motor milestones achieved, and the number of *SMN2* copies[1]. Although SMA is primarily a motor neuron disease[3,9], accumulating evidence points to multiorgan and metabolic abnormalities, especially in the most severe forms[10–19]. Accordingly, developmental and functional defects associated with SMN deficiency have been reported in the autonomic nervous system, liver, lung, intestine, heart, vasculature, bone, endocrine glands[1,10,11,13,15,19–24] and, more recently, in the immune system[25–27] of SMA patients and animal models.

Since *SMN2* copy number represents the primary genetic modifier of clinical disease severity, recent therapeutic approaches have focused on enhancing SMN protein expression by regulating *SMN2* exon 7 splicing processes or by SMN1 replacement[28–30]. Among the SMN-inducing drugs available for SMA therapy[1,31], Nusinersen (Spinraza®) is the first drug approved in infants and adults and the only treatment administered intrathecally[32]. Nusinersen is an 18 base-long 2′-O-(2-methoxyethyl) phosphorothioate antisense oligonucleotide targeting a splicing silencer in the intron 7 of *SMN2* pre-mRNA to promote full-length SMN protein expression[3,20,33]. In infantile-onset SMA1 and SMA2 types, Nusinersen provides a remarkable clinical benefit when administered pre-symptomatically or at the early disease stage. In contrast, its efficacy is far more limited when administered at later stages[32,34–37]. Coherently with the impaired prenatal motor neuron development of severe SMA patients and mouse models[9,20], these observations strongly suggest that the intervention time is the primary factor affecting the clinical outcome of SMN-inducing drugs.

Despite the recent advancement in SMA treatment, there is still an unmet need complemented by SMN-independent approaches that improve clinical outcomes and enhance the benefit of SMN-inducing therapies[20,33,38,39]. Unfortunately, a detailed characterization of biochemical modifications produced by SMN-inducing drugs in the peripheral organs and central nervous system (CNS) remains elusive[40,41]. Therefore, identifying selective metabolic and neurochemical signatures that functionally correlate with disease severity and accurately reflect clinical outcome achievement or lack thereof by SMA therapies is instrumental in guiding the development of new combinatorial therapeutics. Specifically, proteomic and metabolomic investigations focused on the cerebrospinal fluid (CSF) of SMA patients provide the unique opportunity to identify central biochemical disease-associated perturbations and SMA therapies-induced changes[42–46], both of which are unclear.

Using untargeted ¹H-NMR spectroscopy-based metabolomic analysis in the CSF of pediatric SMA patients, we recently unveiled that Nusinersen induces significant changes in energy-related pathways and amino acid metabolism, with a prominent effect on aromatic amino acids (ArAAs)[42]. Beyond their involvement in protein synthesis, hepatic gluconeogenesis and ketogenesis, and hormone synthesis, ArAAs represent the precursors for the monoamine neurotransmitters dopamine (DA), norepinephrine (NE), and serotonin (5-HT)[47,48]. Thus, the reported Nusinersen-related ArAA variations in the CSF of SMA patients might reflect, to some extent, alterations in monoamine neurotransmitter metabolism. This idea is also supported by a study in SMA patients-derived induced pluripotent stem cells (iPSCs) showing significantly reduced gene expression of the aromatic L-amino acid decarboxylase (AADC) enzyme[49].

Considering the outstanding relevance of this issue, in the present work, we used an untargeted NMR-based approach to investigate whether SMN protein deficiency perturbs liver and cerebral metabolome in the SMA mouse model, *SMNΔ7*[50], at different stages of the symptomatology. Afterward, through high-performance liquid chromatography (HPLC), quantitative RT-PCR (qRT-PCR), western blotting (WB), and immunohistochemistry (IHC), we specifically explored the consequences of SMN deficiency in influencing DA, NE, and 5-HT metabolism of *SMNΔ7* mice. Finally, we assessed whether SMN upregulation brought by Nusinersen administration modulates monoamine neurotransmitter levels in the CSF of SMA1, SMA2, and SMA3 patients at therapy loading and maintenance doses relative to baseline.

## Results

### Untargeted NMR-based analysis in *SMNΔ7* mouse liver shows metabolic alterations emerging since the neonatal phase.

Considering the pivotal role of the liver in orchestrating systemic metabolism and energy homeostasis[51], we used untargeted ¹H-NMR spectroscopy-based analysis to explore the consequences of SMN deficiency on the regulation of the hepatic metabolome of *SMNΔ7* mice at both early (post-natal day 3, P3) and late (P11) stages of the disease, compared to age-matched wild-type (WT) littermates.

Using the Chenomx NMR-Suite[52], we identified 44 and 51 metabolites in each 1D ¹H NOESY NMR spectrum of P3 (Fig. 1a) and P11 (Fig. 2a) liver polar extracts, respectively.

Interestingly, Partial Least Square Discriminant Analysis (PLS-DA) indicated that the metabolomic profiles of the P3 and P11 *SMNΔ7* liver were significantly different from that of the respective age-matched WT controls (Fig. 1b; Fig. 2b). Based on the Variable Importance in the Projection (VIP) score analysis[53], we identified ascorbate, succinate, trimethylamine, glutamine, malate, methionine, fumarate, adenosine 3′,5′-diphosphate, 2-aminoadipate, ATP, sarcosine and tyrosine as the discriminating molecules responsible for *SMNΔ7* and WT liver polar extract separation at P3 (Fig. 1c). Evaluation of the magnitude of metabolite changes by hierarchical heatmap and volcano plot revealed that these molecule levels were reduced in *SMNΔ7* mice at early post-natal phase, except for glutamine, whose levels were slightly increased (Fig. 1e, f). On the other hand, NADH, ornithine, succinate, fumarate, acetoacetate, tyrosine, phenylalanine, isoleucine, and malonate represented the molecules responsible for genotype-dependent liver metabolome separation in P11 mice (Fig. 2c). Also in this case, hierarchical heatmap and volcano plot revealed a reduction of these molecules levels in *SMNΔ7* mice at late-symptomatic phase (Fig. 2e–g).

Then, we performed Pathway Enrichment analysis to interpret the NMR-based metabolomic data and identify the specific biochemical pathway alterations linked to SMN deficiency in the liver. At P3, we evidenced significant dysregulation in pyrimidine, purine, urea cycle, and various amino acid metabolism (Fig. 1d;

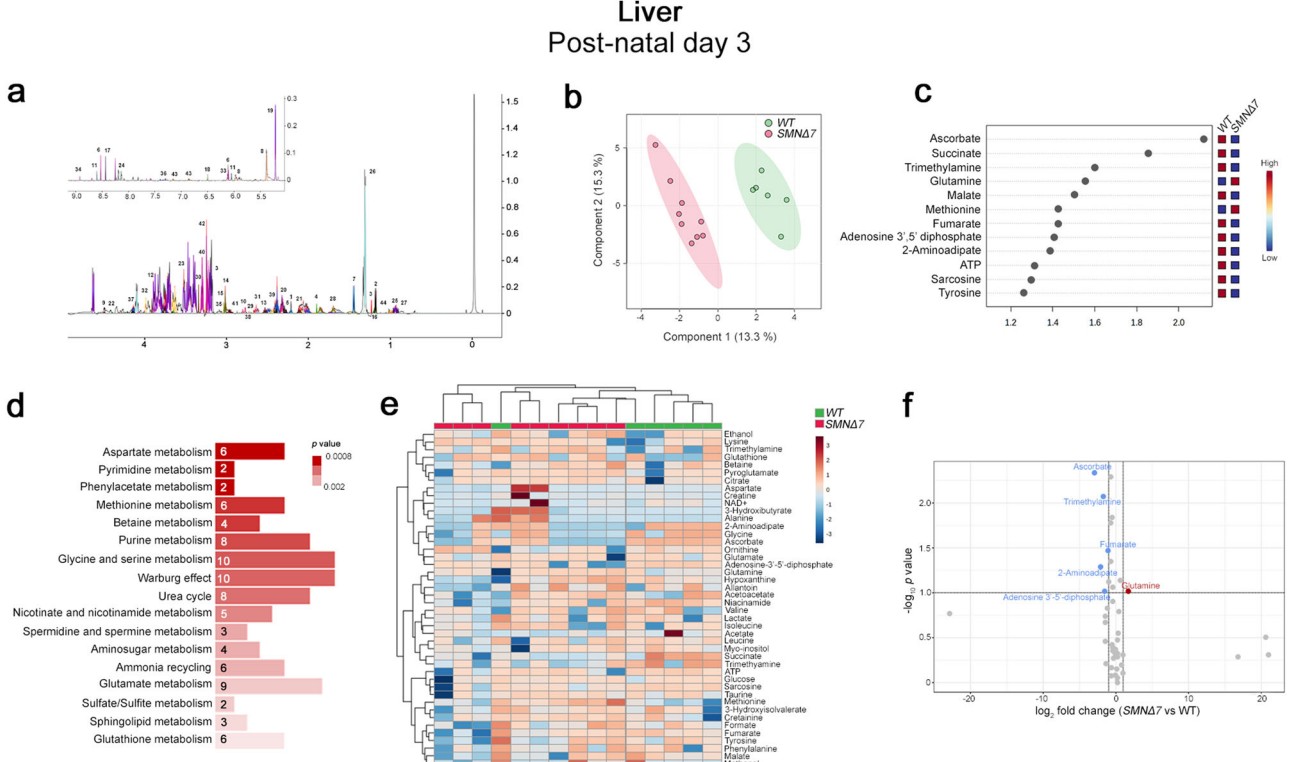

**Fig. 1 Untargeted NMR-based metabolome analysis in the SMNΔ7 mouse liver at post-natal day 3. a** Representative 1D $^1$H NOESY spectrum of liver polar extracts at post-natal day 3 (P3). The spectrum is acquired at 600 MHz and T = 310 K. Fourty-four metabolites are identified and annotated as follows: 1: 2-Aminoadipate; 2: 3-Hydroxybutyrate; 3: 3-Hydroxyisovalerate; 4: Acetate; 5: Acetoacetate; 6: Adenosine 3',5'-diphosphate; 7: Alanine; 8: Allantoin; 9: Ascorbate; 10: Aspartate; 11: ATP; 12: Betaine; 13: Citrate; 14: Creatine; 15: Creatinine; 16: Ethanol; 17: Formate; 18: Fumarate; 19: Glucose; 20: Glutamate; 21: Glutamine; 22: Glutathione; 23: Glycine; 24: Hypoxanthine; 25: Isoleucine; 26: Lactate; 27: Leucine; 28: Lysine; 29: Malate; 30: Methanol; 31: Methionine; 32: myo-Inositol; 33: NAD; 34: Niacinamide; 35: Ornithine; 36: Phenylalanine; 37: Pyroglutamate; 38: Sarcosine; 39: Succinate; 40: Taurine; 41: Trimethylamine; 42: Trimethylamine N-oxide; 43: Tyrosine; 44: Valine **b** PLS-DA score scatter plots showing the metabolomic profile of liver polar extract from *SMNΔ7* (n = 9) and wild-type (WT) (n = 6) mice at P3. The cluster analyses are reported in the Cartesian space that is described by the main components PC1:13.3% and PC2:15.3%. PLS-DA was evaluated using cross-validation (CV) analysis. CV tests performed according to PLS-DA statistical protocol show a significant clusters separation (1.0 accuracy values on both PC1 and PC2, respectively, with positive 0.43 and 0.50 Q2 indices). **c** VIP score graphs of metabolites discriminating *SMNΔ7* from WT livers at P3. Metabolites characterized by a VIP score >1.2 are shown. **d** Diagram of the Pathway Enrichment analysis. The number of molecules (hits) related to the specific metabolic pathway is shown within each bar (see also Supplementary Table 1). **e** Heatmap of changed metabolites. The color of each section corresponds to a concentration value of each metabolite calculated by a normalized concentration matrix (red, upregulated; blue, downregulated). **f** Volcano plot analysis of metabolic changes in *SMNΔ7* and WT livers. Each point on the volcano plot was based on both *p-value* and fold-change values, set at 0.05 and 2.0, respectively. Red points identify up-regulated metabolites whereas blue points identify down-regulated metabolites.

Supplementary Table 1). In addition, genotype-related differences in aerobic glycolysis (Warburg effect) emerged (Fig. 1d; Supplementary Table 1). At P11, we confirmed the occurrence of marked metabolomic differences between *SMNΔ7* mice and WT controls (Fig. 2b), which were mainly addressed to altered mitochondrial-related energy pathways and amino acid dysmetabolism, specifically involving ArAAs and branched-chain amino acids (BCAAs) (Fig. 2c, d; Supplementary Table 2).

The present data in *SMNΔ7* mice put forward a harmful influence of low SMN protein levels in disrupting hepatic metabolism since the early post-natal stage.

**Untargeted NMR-based analysis in *SMNΔ7* brain unveils energy-related pathways and amino acids metabolism alteration only at the symptomatic stage.** Although previous observations reported a role for SMN in regulating brain development in animal models[54] and patients[55], the specific influence of SMN deficiency on the alteration of cerebral metabolome has not been fully elucidated. To clarify this question, we extended the

NMR-based analysis to the brain of *SMNΔ7* mice, relative to their age-matched WT controls.

At both P3 and P11, 22 metabolites were measured in each 1D $^1$H NOESY NMR spectrum according to qualitative and quantitative analysis of polar brain extracts performed with Chenomix NMR-Suite (Fig. 3a; Supplementary Fig. 1).

In contrast with the early genotype-dependent differences found in the liver (Fig. 1), the PLS-DA showed that the metabolomic profile of *SMNΔ7* brains is indistinguishable from that of WT brains at P3 (Supplementary Fig. 1). Conversely, PLS-DA indicated significant variations in the cerebral metabolome of overt symptomatic *SMNΔ7* mice compared to age-matched WT littermates (Fig. 3b). Specifically, VIP score analysis identified tyrosine as the most significant classifier between genotypes (Fig. 3c). Succinate, inosine monophosphate (IMP), acetate, and threonine appeared as further discriminating metabolites responsible for genotype-dependent metabolome separation (Fig. 3c). Interestingly, pathway enrichment analysis unraveled alterations in pyruvate metabolism, BCAA degradation, glycine and serine metabolism, and ArAA metabolism in the brain of *SMNΔ7* mice, compared to control

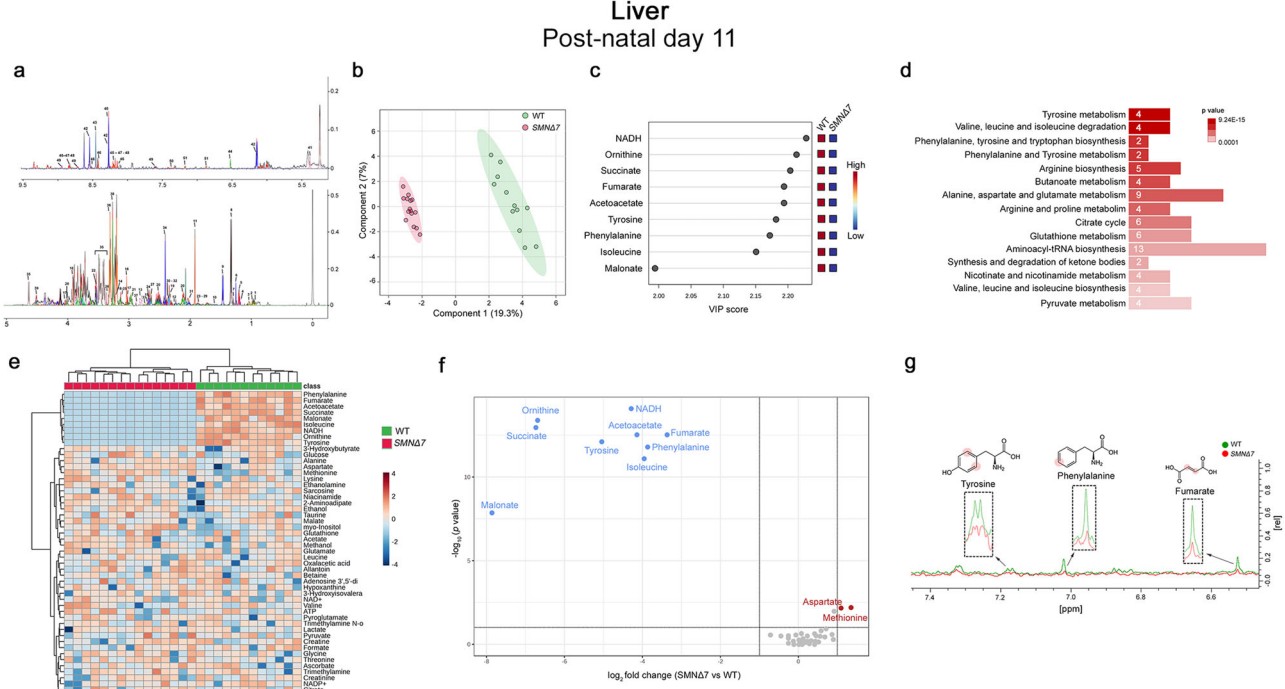

**Fig. 2 Untargeted NMR-based metabolome analysis in the *SMNΔ7* mouse liver at post-natal day 11. a** Representative 1D CPMG spectrum of liver polar extracts at post-natal day 11 (P11). The spectrum is acquired at 600 MHz and T = 310 K. Fifty-one metabolites are identified and annotated as follows: 1: Isoleucine; 2: Leucine; 3: Valine; 4: Ethanol; 5: 3-Hydroxybutyrate; 6: 3-Hydroxyisovalerate; 7: Threonine; 8: Lactate; 9: Alanine; 10: 2-Aminoadipate; 11: Acetate; 12: Acetoacetate; 13: Aspartate; 14: Betaine; 15: Citrate; 16: Creatine; 17: Creatinine; 18: Ethanolamine; 19: Glutamate; 20: Glutamine; 21: Glutathione; 22: Glycine; 23: Lysine; 24: Malate; 25: Malonate; 26: Methanol; 27: Methionine; 28: Myo-inositol; 29: Ornithine; 30: Oxalacetic acid; 31: Pyroglutamate; 32: Pyruvate; 33: Sarcosine; 34: Succinate; 35: Glucose; 36: Taurine; 37: Trimethylamine; 38: Trimethylamine N-oxide; 39: Ascorbate; 40: Adenosine 3′,5′-diphosphate; 41: Allantoin; 42: ATP; 43: Formate; 44: Fumarate; 45: Hypoxanthine; 46: NADH; 47: NAD + ; 48: NADP + ; 49: Niacinamide; 50: Phenylalanine; 51: Tyrosine. **b** PLS-DA score scatter plots showing the metabolomic profile of liver polar extract from *SMNΔ7* (n = 15) and wild-type (WT) (n = 12) mice. The cluster analyses are reported in the Cartesian space that is described by the main components PC1:19.3% and PC2:7%. PLS-DA was evaluated using cross-validation (CV) analysis. CV tests performed according to PLS-DA statistical protocol show a significant clusters separation (1.0 accuracy values on both PC1 and PC2, respectively, with positive 0.91 and 0.94 Q2 indices). **c** VIP score graph of metabolites discriminating cluster. Metabolites characterized by a VIP score > 1.4 are shown. **d** Diagram of the Pathway Enrichment analysis. The number of molecules (hits) related to the specific metabolic pathway is shown within each bar (see also Supplementary Table 1). **e** Heatmap of changed metabolites. The color of each section corresponds to a concentration value of each metabolite calculated by a normalized concentration matrix (red, upregulated; blue, downregulated). **f** Volcano plot analysis of metabolic changes in *SMNΔ7* and WT polar liver extracts. Each point on the volcano plot was based on both *p*-value and fold-change values, set at 0.05 and 2.0, respectively. Red points identify up-regulated metabolites whereas blue points identify down-regulated metabolites. **g** Superimposition of 1D ¹H-NMR spectra of *SMNΔ7* (red) and WT (green) polar liver extracts. The dashed boxes show the zooms of discriminating metabolites in the superimposition. The proton signals displayed in the overlayed spectra are highlighted by red dots in the respective structures of each metabolite.

littermates (Fig. 3d; Supplementary Table 3). Accordingly, hierarchical heatmap and volcano plot analyses evidenced greater cerebral levels of tyrosine, coupled with lower amounts of succinate, IMP, threonine, and 3-hydroxyisovalerate in the late-symptomatic *SMNΔ7* brains, compared with WT controls (Fig. 3 e–g).

Our untargeted metabolomic analysis in the brain of *SMNΔ7* mice reveals a striking influence of SMN depletion in perturbing cerebral amino acid metabolism and energy homeostasis pathways only at the overt symptomatic stage of the disease following the occurrence of morphological brain alterations[54].

**Decreased norepinephrine levels in the brain and spinal cord of late-symptomatic *SMNΔ7* mice.** In light of the notable alterations in ArAA metabolism observed in *SMNΔ7* mice, we investigated whether SMN deficiency affects monoamine neurotransmitter levels in the CNS of SMA mice compared to age-matched WT controls.

To disentangle this issue, we used HPLC coupled with electrochemical detection to directly measure the levels of DA,

NE, 5-HT and their main precursors and catabolites, including L-dihydroxyphenylalanine (L-DOPA), dihydroxyphenylacetic acid (DOPAC), homovanillic acid (HVA) and 5-hydroxy-indol acetic acid (5-HIAA), in the brain and spinal cord of *SMNΔ7* mice at progressive symptomatology stages (P3, P6 and P11)[50], and their age-matched control littermates (Fig. 4a, e).

Statistical analysis unveiled a significant age-dependent difference in brain DA levels between genotypes, with *SMNΔ7* mice showing a slightly higher content of this catecholamine at the late-symptomatic stage compared to age-matched control animals (two-way ANOVA: age, $F_{(2,51)} = 132.3$, $p < 0.0001$; age×genotype, $F_{(2,51)} = 8.356$, $p = 0.0007$; P11: *SMNΔ7* *vs* WT, $p < 0.0001$, multiple comparisons controlled by Benjamini and Hochberg False Discovery Rate method; Fig. 4b, Supplementary Table 4).

In contrast to its precursor, we found a remarkable reduction of cerebral NE levels in *SMNΔ7* mice at both P6 and P11, compared to their respective age-matched controls (genotype, $F_{(1,51)} = 15.71$, $p = 0.0002$; age×genotype, $F_{(2,51)} = 3.247$, $p = 0.0471$; P6, $p = 0.0002$; P11, $p = 0.0133$; Fig. 4c; Supplementary Table 4). Conversely, we did not detect significant differences in cerebral

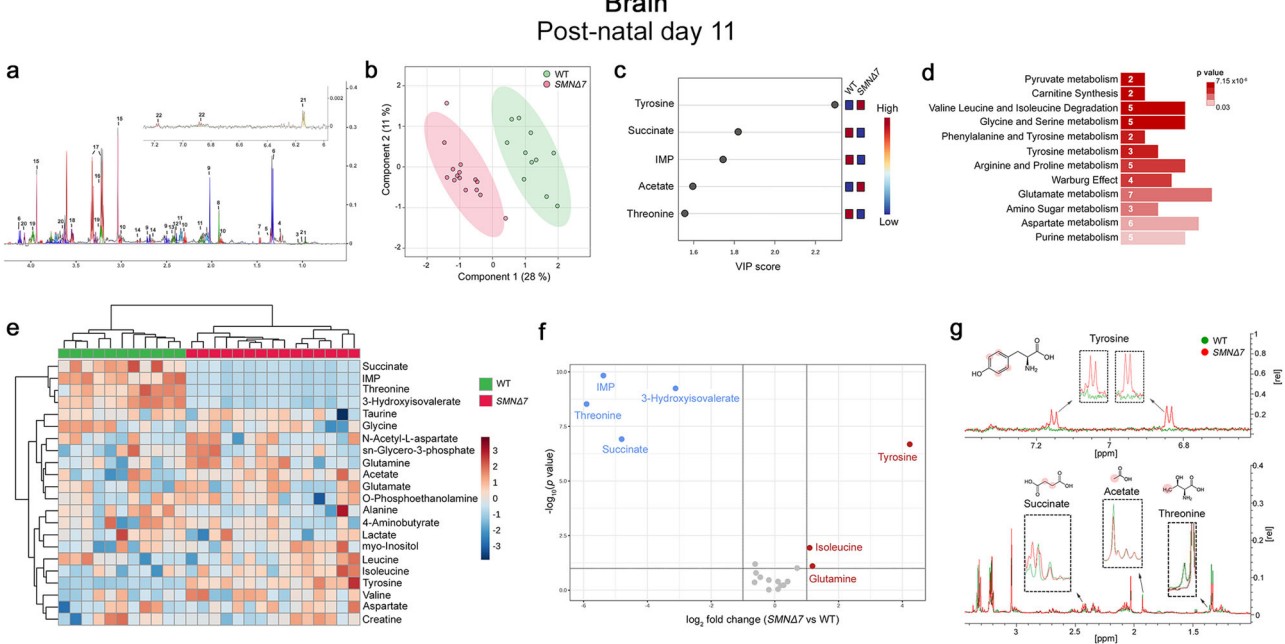

**Fig. 3 Untargeted NMR-based metabolome analysis in the *SMNΔ7* mouse brain at post-natal day 11. a** Representative 1D $^1$H NOESY spectrum of mouse brain extracts at post-natal day 11 (P11). The spectrum is acquired at 600 MHz and T = 310 K. Twenty-two polar metabolites are annotated as follows: 1: Leucine; 2: Isoleucine; 3:Valine; 4: 3-Hydroxyisovalerate; 5: Threonine; 6: Lactate; 7: Alanine; 8: Acetate; 9: N-Acetyl-L-aspartate 10: 4-Aminobutyrate; 11: Glutamate; 12: Succinate; 13: Glutamine; 14: Aspartate; 15: Creatine; 16: sn-Glycero-3-phosphocholine; 17: Taurine; 18: Glycine; 19: O-Phosphoethanolamine; 20: Myo-inositol; 21: 5-Inosine-monophosphate; 22: Tyrosine. **b** PLS-DA score scatter plots showing the metabolomic profile of *SMNΔ7* (n = 15) and wild-type (WT) (n = 11) mouse brain. The cluster analyses are reported in the Cartesian space that is described by the main components PC1:28% and PC2:11%. PLS-DA was evaluated using cross-validation (CV) analysis. CV tests performed according to PLS-DA statistical protocol show significant separation clusters (0.94 and 1.0 accuracy values on PC1 and PC2, respectively, with positive 0.70 and 0.80 Q2 indices). **c** VIP score graphs of the metabolites discriminating the cerebral profile of *SMNΔ7* from WT mice at P11. Metabolites characterized by a VIP score > 1.4 are shown. **d** Diagram of the Pathway Enrichment analysis. The number of molecules (hits) related to the specific metabolic pathway is shown within each bar (see also Supplementary Table 2). **e** Heatmap of changed metabolites. The color of each section corresponds to a concentration value of each metabolite calculated by a normalized concentration matrix (red, up-regulated; blue, downregulated). **f** Volcano plot analysis of metabolic changes in *SMNΔ7* and WT brain polar extracts at P11. Each point on the volcano plot was based on both *p-value* and fold-change values, set at 0.05 and 2.0, respectively. Red points identify up-regulated metabolites whereas blue points identify down-regulated metabolites. **g** Superimposition of 1D $^1$H-NMR spectra of *SMNΔ7* (red) and WT (green) polar brain extracts. The dashed boxes show the zooms of discriminating metabolites in the superimposition. The proton signals displayed in the overlayed spectra are highlighted by red dots in the respective structures of each metabolite.

5-HT levels between genotypes (Fig. 4d; Supplementary Table 4), although a dramatic increase in the main 5-HT catabolite, 5-HIAA, was found in *SMNΔ7* mice at both P6 and P11, compared to the respective age-matched WT mice (Supplementary Fig. 2, Supplementary Table 4). Yet, we did not observe any significant variation in the L-DOPA, DOPAC, and HVA levels between *SMNΔ7* and control brains at any time-point tested (Supplementary Fig. 2; Supplementary Table 4).

Notably, we confirmed significantly decreased NE levels in the spinal cord of *SMNΔ7* mice at P6 and showed a trend towards reduction of this neurotransmitter levels at P3, compared to the age-matched WT littermates (genotype, $F_{(1,49)} = 12.31$, $p = 0.0010$; age×genotype, $F_{(2,49)} = 2.923$, $p = 0.0632$; P3, $p = 0.0938$; P6, $p = 0.0003$; Fig. 4g; Supplementary Table 4). Comparable DA and 5-HT levels were found between genotypes at all stages evaluated (Fig. 4f, h; Supplementary Table 4). As reported for the brain, there was a significant increase in 5-HIAA levels and 5-HIAA/5-HT ratio in the *SMNΔ7* spinal cord at the late-symptomatic stage of the disease compared to age-matched WT controls (Supplementary Fig. 2; Supplementary Table 4). Conversely, unaltered levels of the other monoamine-related metabolites were found between genotypes (Supplementary Fig. 2; Supplementary Table 4).

Our neurochemical results unveiled a striking decrease in NE levels in the brain and, to a lesser extent, in the spinal cord of *SMNΔ7* mice, suggesting that SMN deficiency triggers deleterious effects on catecholamine neurotransmission in an age-dependent manner. In addition, the present HPLC data also showed a considerable alteration in 5-HT turnover in the CNS of SMA mice.

**SMN deficiency affects the expression of monoamine-regulating enzymes in the *SMNΔ7* brain.** Next, to identify the molecular determinants underlying the neurochemical abnormalities found in the *SMNΔ7* brain, we investigated whether SMN deficiency perturbs the expression of the main enzymes regulating DA, NE, and 5-HT levels during post-natal ontogeny. First, we performed WB analyses in the brains of P3, P6, and P11 *SMNΔ7* mice and age-matched WT controls (Fig. 5).

We measured the expression levels of the rate-limiting enzyme for DA production, tyrosine hydroxylase (TH), which converts tyrosine into the DA precursor, L-DOPA. We found comparable TH protein levels in *SMNΔ7* and WT brains throughout the ages analyzed (Fig. 5a, b, h, i, o, p; Supplementary Fig. 10). Conversely, a significant increase in P-Ser40-TH levels was identified in the *SMNΔ7* brain at P11 compared to age-matched WT animals ($p = 0.035$; unpaired *t*-test; Fig. 5a, c, h, j, o, q; Supplementary Fig. 10). Then, we analyzed the expression of dopamine β-hydroxylase (DβH), the rate-limiting enzyme for NE production,

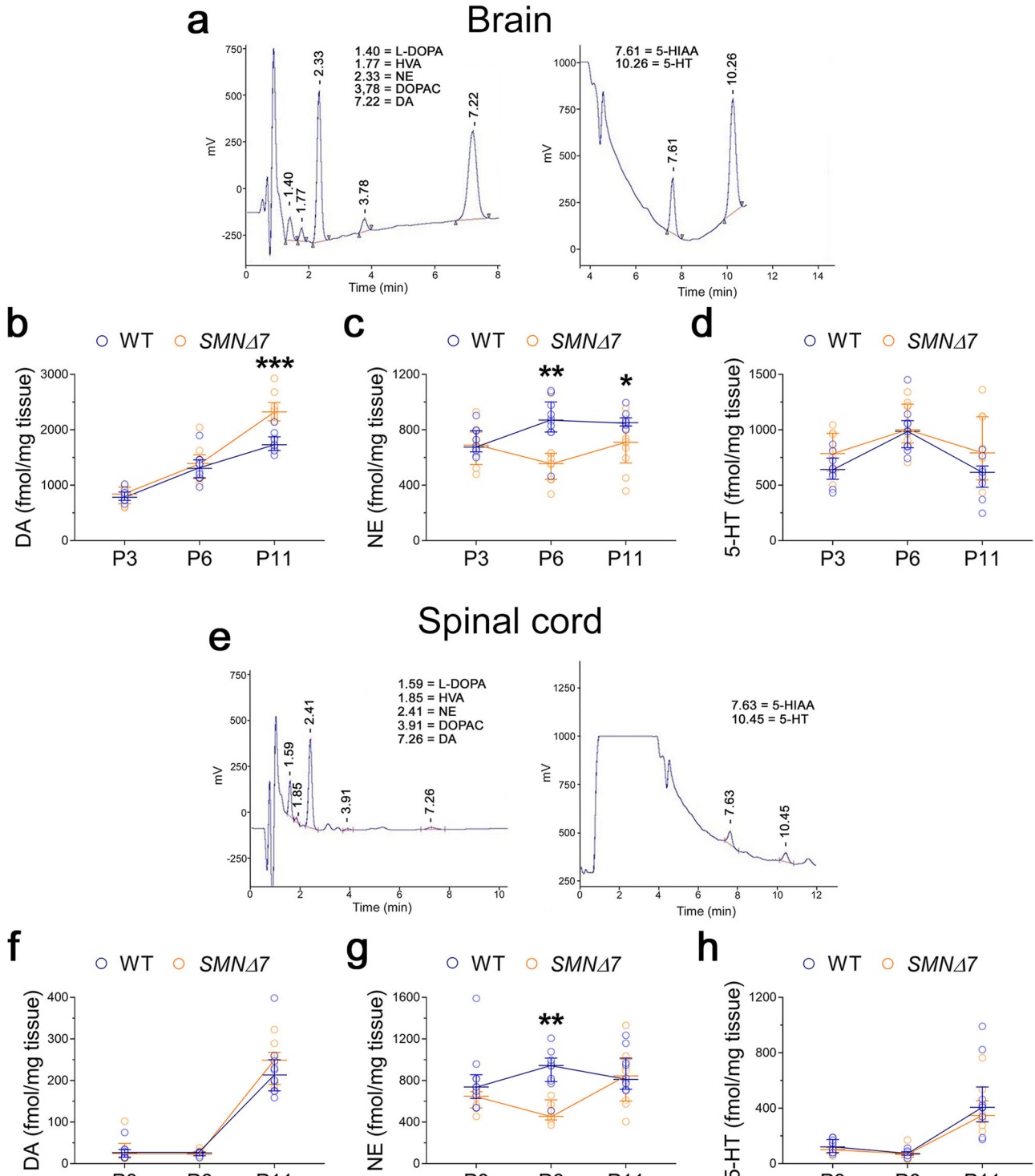

**Fig. 4 Dopamine, norepinephrine, and serotonin levels in the brain of *SMNΔ7* mice. a** Representative HPLC chromatograms of the *SMNΔ7* mouse brain. **b–d** Levels of **b** dopamine (DA), **c** norepinephrine (NE), and **d** serotonin (5-HT) in the brain of *SMNΔ7* mice and their matched wild type (WT) at post-natal day 3 (P3, n = 10 WT, n = 8 *SMNΔ7*), P6 (n = 10 WT, n = 9 *SMNΔ7*) and P11 (n = 10/genotype). **e** Representative HPLC chromatograms of the *SMNΔ7* mouse spinal cord. **f–h** Levels of **f** DA, **g** NE, and **h** 5-HT in the spinal cord of *SMNΔ7* mice at P3 (n = 10 WT, n = 6 *SMNΔ7*), P6 (n = 10 WT, n = 9 *SMNΔ7*) and P11 (n = 10/genotype). The average amount of each monoamine detected was normalized for mg of wet tissue. Data are expressed as median with interquartile range (IQR) and analyzed by two-way ANOVA, followed by multiple comparisons controlled by False Discovery Rate method of Benjamini and Hochberg. *$p < 0.05$, **$p < 0.01$, ***$p < 0.0001$, compared to age-matched WT mice.

which converts DA into NE within noradrenergic neurons of the *locus coeruleus* (LC)[56,57]. Consistent with a significant decrease in cerebral NE levels at P6 and P11 (Fig. 4c), we found a prominent reduction of DβH levels in the brain of *SMNΔ7* mice at both P6

and P11, compared to age-matched controls (P6, $p = 0.038$; P11, $p = 0.037$; Fig. 5a, d, h, k, o, r; Supplementary Fig. 10). We also disclosed a significant reduction in AADC protein levels in the late-symptomatic *SMNΔ7* mouse brain compared with age-matched

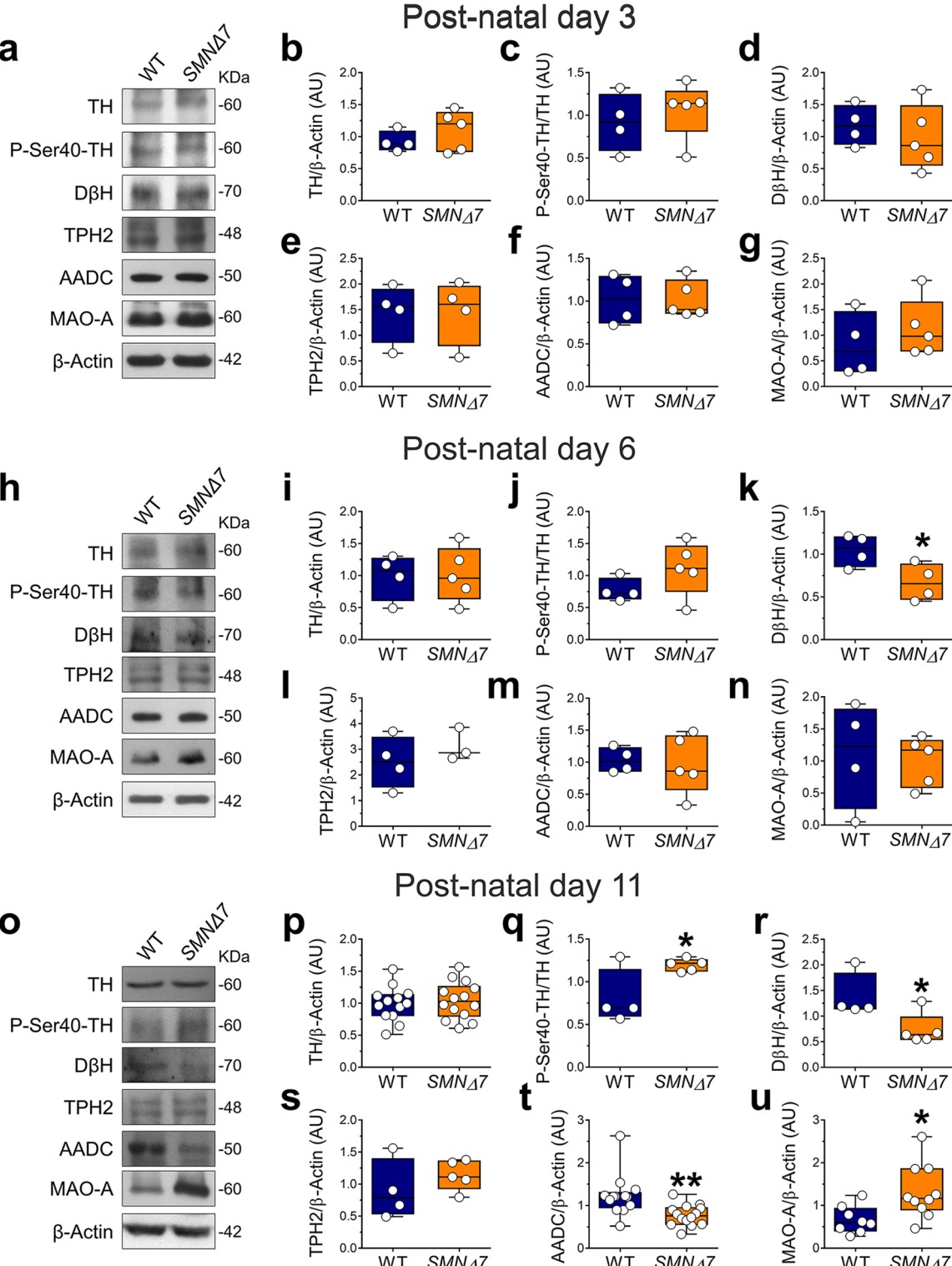

controls ($p = 0.004$; Fig. 5a, f, h, m, o, t; Supplementary Fig. 10). Differently from DβH and AADC, comparable tryptophan hydroxylase 2 (TPH2) levels were reported between genotypes at any age tested (Fig. 5a, e, h, l, o, s; Supplementary Fig. 10). Interestingly, WB experiments showed a significant upregulation of the major monoamine-degrading enzyme, monoamine oxidase A (MAO-A)[58], in late-symptomatic *SMNΔ7* mice compared to

age-matched controls ($p = 0.016$; Fig. 5a, g, h, n, o, u; Supplementary Fig. 10). On the other hand, comparable cerebral protein levels of MAO-B, catechol-*O*-methyltransferase (COMT) (Supplementary Fig. 3; Supplementary Fig. 11) and phenylalanine hydroxylase (PAH) (Supplementary Fig. 4; Supplementary Fig. 12) were found between genotypes at any age tested. Similarly, we found unaltered levels of the monoamine-regulating enzymes

**Fig. 5 Expression levels of monoamine-regulating enzymes in the brain of *SMNΔ7* mice. a, h, o** Representative autoradiograms of brain lysates immunoblots of *SMNΔ7* and wild-type (WT) mice at **a** post-natal day 3, **h** day 6, and **o** day 11. **b–g** Protein levels quantification of **b** Tyrosine hydroxylase (TH), **c** phospho-Tyrosine hydroxylase at Ser-40 (P-Ser40-TH), **d** Dopamine β hydroxylase (DβH), **e** Tryptophan hydroxylase 2 (TPH2), **f** Aromatic amino acid decarboxylase (AADC) and **g** Monoamine Oxidase A (MAO-A) in *SMNΔ7* (n = 4) and WT (n = 5) mice (except for TPH2: n = 4 mice/genotype) at post-natal day 3. **i–n** Protein levels quantification of **i** TH (n = 4 WT, n = 5 *SMNΔ7*), **j** P-Ser40-TH (n = 4 WT, n = 5 *SMNΔ7*), **k** DβH (n = 4 WT, n = 4 *SMNΔ7*), **l** TPH2 (n = 4 WT, n = 3 *SMNΔ7*), **m** AADC (n = 4 WT, n = 5 *SMNΔ7*) and **n** MAO-A (n = 4 WT, n = 5 *SMNΔ7*) at post-natal day 6. **p–u** Protein levels quantification of **p** TH (n = 13 WT, n = 14 *SMNΔ7*), **q** P-Ser40-TH (n = 4 WT, n = 5 *SMNΔ7*), **r** DβH (n = 4 WT, n = 5 *SMNΔ7*), **s** TPH2 (n = 4 WT, n = 5 *SMNΔ7*), **t** AADC (n = 13 WT, n = 14 *SMNΔ7*) and **u** MAO-A (n = 8 WT, n = 10 *SMNΔ7*) at post-natal day 11. Data are normalized to β-Actin levels and shown as box and whisker plots representing the median with interquartile range (IQR). Dots represent individual mice values. *$p < 0.05$, **$p < 0.01$, compared with age-matched WT mice (unpaired *t*-test).

(Supplementary Fig. 5; Supplementary Fig. 13) and PAH (Supplementary Fig. 4; Supplementary Fig. 12) in the spinal cord of *SMNΔ7* compared with WT littermates during the ontogeny.

Our findings reveal an age-dependent influence of SMN deficiency in dysregulating the protein levels of critical enzymes involved in monoamine neurotransmitter metabolism, corroborating at a molecular level the neurochemical abnormalities found in SMA mice.

**SMN deficiency affects the expression of the genes regulating monoamine metabolism in the *SMNΔ7* mouse brain.** We then investigated the effect of SMN deficiency on regulating the mRNA levels of genes encoding the enzymes involved in monoamine metabolism. Accordingly, we performed qRT-PCR analysis in the brain of *SMNΔ7* and WT mice at P3 and P11. In line with the protein expression data, we found a significant down-regulation of both *Dβh* and *Aadc* mRNA levels in *SMNΔ7* brains at P11 but not at P3, compared to their respective age-matched WT controls (*Dβh: p = 0.002; Aadc: p = 0.0492*; Fig. 6b, d; Supplementary Fig. 6). Comparable *Th, Tph2, Mao-b,* and *Comt* gene expression was found among genotypes at P11 (Fig. 6a, c; Supplementary Fig. 7). Surprisingly, despite the increased MAO-A protein levels found in *SMNΔ7* brain at P11 (Fig. 4), qRT-PCR data showed lower *Mao-a* transcript levels compared to controls (*p = 0.009*; Supplementary Fig. 7).

Furthermore, we investigated whether the abnormally lower *Dβh* and *Aadc* transcript levels reported in the *SMNΔ7* mouse were also common to other motor neuron diseases, such as amyotrophic lateral sclerosis (ALS). To this aim, we measured the cerebral mRNA levels of *Th, DβH, Tph2, Aadc, Mao-a,* and *Comt* in adult (P120) symptomatic *SOD1* mice, reported as a reliable animal model of ALS[59,60]. Interestingly, qRT-PCR analysis found no significant difference between *SOD1* and control littermates for all transcripts examined (Fig. 6 e–h).

Our findings highlight that the abnormal expression of *Dβh* and *Aadc* depends on SMN deficiency rather than being a nonspecific effect due to motor neuron degeneration.

**Decreased expression of TH and DβH in the *locus coeruleus* of late-symptomatic *SMNΔ7* mice.** To further characterize the influence of SMN deficiency on the regional expression of monoamine-regulating enzymes, we performed quantitative immunohistochemistry (IHC) and stereological cell count within the brainstem monoaminergic *nuclei* of the *substantia nigra pars compacta* (SNc), ventral tegmental area (VTA), *raphe nucleus* and *locus coeruleus* (LC) of *SMNΔ7* and WT mice at P11 (Fig. 7).

We quantified AADC⁺ and TH⁺ neurons in the dopaminergic regions of the *SNc* and VTA (Fig. 7a–c). In line with WB and qRT-PCR experiments carried out in the whole brain, IHC analysis showed a significant reduction of AADC⁺ neurons, both in the SNc and VTA of *SMNΔ7* mice compared to WT littermates (SNc, *p = 0.014*; VTA, *p = 0.046*; unpaired *t*-test; Fig. 7b, c).

Conversely, a comparable TH⁺ cell number was found in the midbrain of *SMNΔ7* and WT mice (Fig. 7b, c). Consistently, we observed a significant reduction of the AADC/TH ratio in both the SNc and VTA of *SMNΔ7* mice compared to controls (SN, *p = 0.0132*; VTA, *p = 0.045*; Fig. 7b, c).

We then measured the number of DβH⁺ and TH⁺ neurons in the LC (Fig. 7d, e). IHC data showed a drastic reduction of DβH⁺ neurons in *SMNΔ7* mice compared to WT littermates (*p = 0.001*; Fig. 7e). Interestingly, we also found a ~35% decrease of TH immunostaining in the LC of *SMNΔ7* mice, compared to WT controls, although not reaching the statistical significance (*p = 0.0581*; Fig. 7e). As a result, the stereological analysis highlighted a reduced DβH/TH ratio in overt symptomatic *SMNΔ7* mice compared to WT controls at P11 (*p = 0.001*; Fig. 7e).

Finally, we measured the number of AADC⁺ and TPH2⁺ neurons in the serotonergic region of the *raphe nucleus* (Fig. 7f, g). As with dopaminergic midbrain neurons, we observed a marked decrease in AADC⁺ neurons in *SMNΔ7* mice compared to controls (*p = 0.0001*; Fig. 7g), mirrored by unaltered TPH2 immunostaining. Accordingly, we found a significant AADC/TPH2 ratio reduction in *SMNΔ7* mice compared to WT littermates (*p < 0.0001*; Fig. 7g).

The prominent downregulation of DβH and AADC expression in the *SMNΔ7* brainstem confirms the detrimental effect of SMN deficiency on monoaminergic neurotransmitter metabolism. On the other hand, the substantial reduction of NE-positive neurons within the LC of *SMNΔ7* mice is in line with the high vulnerability of this brain nucleus to neuronal degeneration under a neuroinflammatory environment typically occurring in SMA pathology[25–27,61]. Accordingly, loss of noradrenergic LC neurons is described in animal models and patients with other neurodegenerative disorders, including multiple sclerosis, Alzheimer's disease, and Parkinson's disease[62–64].

**Untargeted NMR-based metabolome analysis in the CSF of naïve SMA1 patients unveils deregulated aromatic amino acid and energy-related metabolism.** To explore whether SMN deficiency in severe patients produces central metabolomic features encompassing those found in late-symptomatic *SMNΔ7* brain, we analyzed the CSF metabolome profile of a small cohort of naïve SMA1 infants (n = 3) and healthy pediatric controls (n = 3) (Supplementary Table 5).

Thirty-five metabolites were measured in each 1D ¹H NOESY NMR spectrum according to qualitative and quantitative analysis of polar brain extracts performed with Chenomix NMR-Suite (Fig. 8a).

Heatmap and volcano plot analysis highlighted higher CSF levels of the ArAAs, tyrosine, phenylalanine, and tryptophan, and the ketone bodies, 3-hydroxybutyric acid, acetoacetate, and acetone, in SMA1 patients compared to healthy controls (Fig. 8b–d). Moreover, NMR data revealed decreased levels of pyroglutamic acid, 2-hydroxyisovalerate, and creatinine in severe SMA patients (Fig. 8b, c). The reduction of creatinine in the CSF of SMA1 patients further

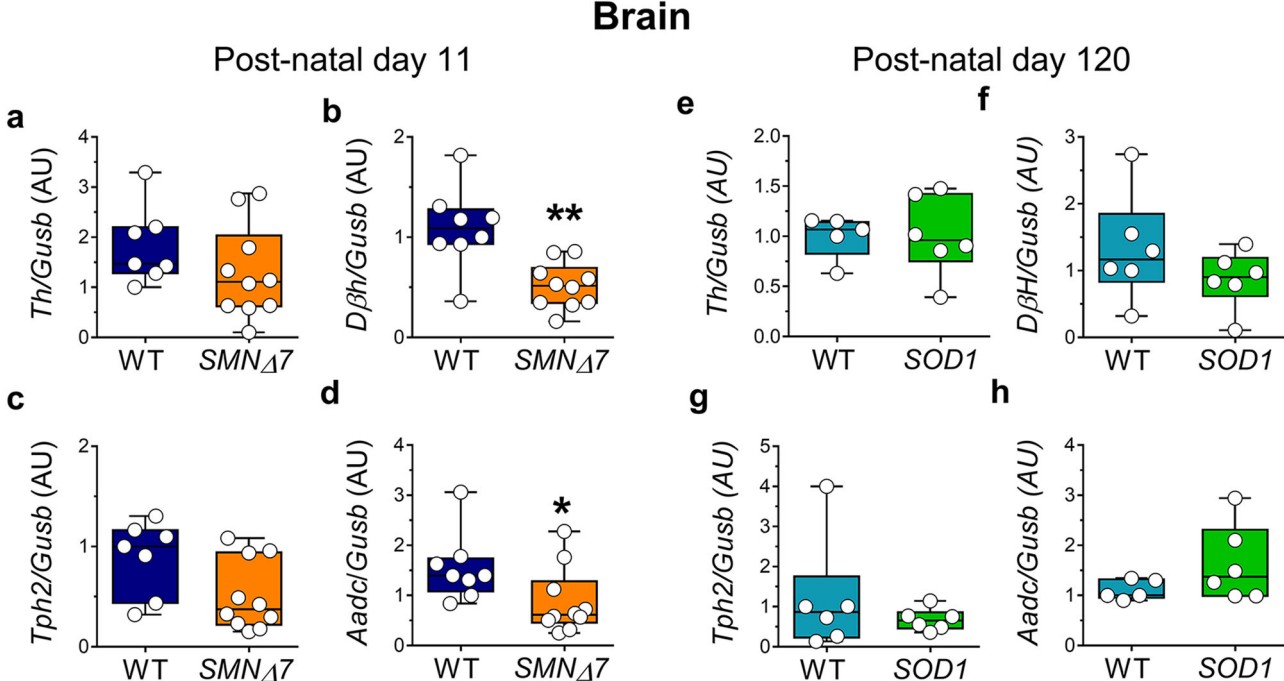

**Brain**

**Post-natal day 11** **Post-natal day 120**

**Fig. 6 Expression levels of the genes encoding monoamine-regulating enzymes in the brain of *SMNΔ7* mice at post-natal day 11. a–d** Transcript levels of **a** *Tyrosine hydroxylase* (*Th;* n = 7 WT, n = 10 *SMNΔ7*), **b** *Dopamine β hydroxylase* (*Dβh;* n = 8 WT, n = 10 *SMNΔ7*), **c** *Tryptophan hydroxylase 2* (*Tph2;* n = 7 WT, n = 10 *SMNΔ7*) and **d** *Aromatic amino acid decarboxylase* (*Aadc;* n = 8 WT, n = 10 *SMNΔ7*) in the brain of *SMNΔ7* mice and wild-type (WT) littermates at post-natal day 11. **e–h** Transcript levels of **e** *Tyrosine hydroxylase* (*Th*), **f** *Dopamine β hydroxylase* (*Dβh*), **g** *Tryptophan hydroxylase 2* (*Tph2*) and **h** *Aromatic amino acid decarboxylase* (*Aadc*) in the brain of *SOD1* and wild-type (WT) littermates at post-natal day 120 (n = 6/genotype). Data are expressed as the difference in threshold cycle ($2^{-\Delta\Delta Ct}$) between the target gene and the reference gene, *β-glucuronidase* (*Gusb*) (arbitrary units, AU), and shown as box and whisker plots representing median with interquartile range (IQR). Dots represent individual mice values. *$p < 0.05$, **$p < 0.01$, compared with age-matched WT mice (unpaired *t*-test).

suggests a role for this molecule as a possible candidate biomarker of neuromuscular degeneration in SMA[42,43,65].

Our NMR findings in the CSF of severe SMA patients highlighted an influence of SMN deficiency on perturbing central ArAAs and BCAAs availability, which is consistent with the metabolomic results observed in the brain of late-symptomatic *SMNΔ7* mice (Fig. 3e–g). In addition, ketone body accumulation in the CSF of SMA1 patients compared to healthy controls aligns with glucose metabolism alterations previously reported in patients with the severe form of the disease[12,13,15,16,18,19,22].

**Nusinersen therapy increases CSF norepinephrine levels in severe but not milder SMA patients.** Finally, we performed a retrospective, longitudinal analysis in a cohort of 33 SMA patients with different disease severity (12 SMA1, 10 SMA2, and 11 SMA3) to assess the effect of Nusinersen treatment on the CSF concentration of NE, 5-HT, and 5-HIAA at different stages of the therapy (Supplementary Table 6). CSF samples were collected at the time of the first Nusinersen treatment (T0, baseline level), as well as at 64 (T1, loading phase) and 302 (T2, maintenance phase) days after the initial treatment and the NE, 5-HT, and 5-HIAA levels measured by HPLC coupled with electrochemical detection (Fig. 9a, b; Supplementary Fig. 8).

Initially, we analyzed all Nusinersen-treated SMA patients as a whole cohort, irrespective of their different disease severity. Statistical analysis showed that the NE, 5-HT, and 5-HIAA content at T1 or T2 was comparable to that of the same patients before Nusinersen therapy (Fig. 9c; Supplementary Fig. 8).

Then, we analyzed SMA patients as separate cohorts to unveil whether Nusinersen treatment exerts disease severity-dependent neurochemical changes. Notably, in SMA1 patients, we reported a

statistically significant increase of CSF NE levels at both T1 and T2, compared to T0 (median [IQR] of nM concentrations: T0 = 9.54 [1.13;14.56] *vs* T1 = 13.91 [12.01;17.45], $p = 0.006$; T0 *vs* T2 = 13.97 [5.58;30.54], $p = 0.005$; Wilcoxon matched-pairs signed ranks test; Fig. 9e; Supplementary Table 7). Importantly, the Nusinersen-dependent NE upregulation reported in SMA1 patients was not associated with age or BMI (Supplementary Table 8). In contrast to SMA1, in milder SMA2 and SMA3 patients, we did not find any significant modulatory effect of Nusinersen treatment on the central NE concentrations at any treatment stage analyzed (Fig. 9h, k; Supplementary Table 7).

Next, we evaluated whether the increased NE levels in the CSF of SMA1 patients at T2 were functionally associated with clinical motor outcome (Supplementary Table 9). To this aim, we stratified SMA1 patients according to their motor response to the therapy. Based on previous reports[66,67], we defined a motor milestone achievement of Nusinersen-treated SMA1 patients as an increase of at least 4 points in the CHOP-INTEND scale at T2 relative to T0 (Fig. 9d). Spaghetti plot showed that each Nusinersen-treated SMA1 patient presented higher NE levels compared to the respective naïve condition, regardless of the CHOP-INTEND improvement achieved (Fig. 9f; Supplementary Table 8). Similarly, in SMA2 and SMA3 patients we did not find any NE variation associated with Nusinersen-mediated improvement in motor outcomes, assessed by HFMSE score[68] (Fig. 9g, j, i, l; Supplementary Fig. 8; Supplementary Tables 7 and 8).

Notably, HPLC analysis showed no main changes in 5-HT and 5-HIAA levels in the CSF of Nusinersen-treated SMA1 patients, at both T1 and T2, compared with T0 (Supplementary Fig. 8; Supplementary Table 7). Furthermore, we excluded specific associations of 5-HT and 5-HIAA levels with the age or BMI of

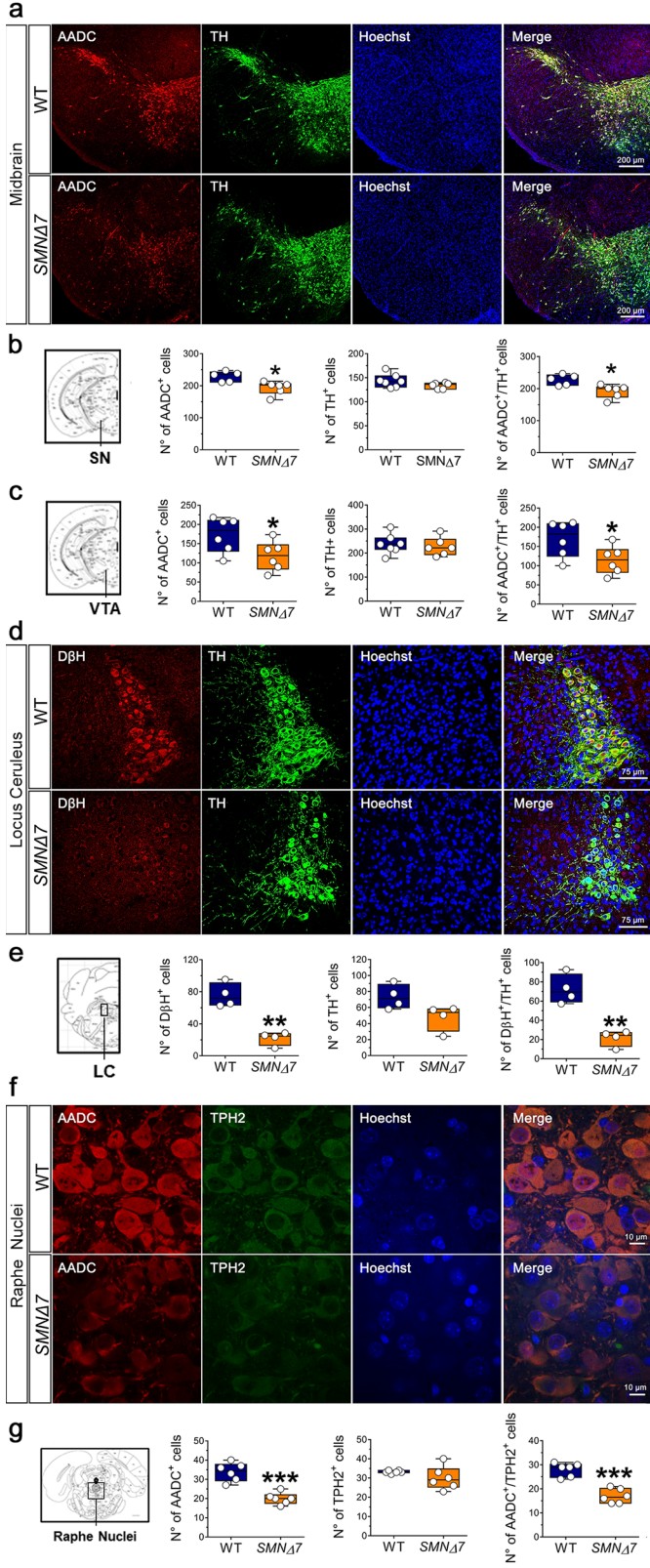

**Fig. 7 Immunohistochemical analysis of monoamine-regulating enzymes in the brainstem monoaminergic nuclei of *SMNΔ7* mice at post-natal day 11. a** Confocal images showing the labeling of AADC (red) and TH (green) in the *substantia nigra pars compacta (SNc)* and ventral tegmental area (VTA) of *SMNΔ7* and wild-type (WT) mice at post-natal day 11. Nuclei were counterstained with Hoechst (blue). Merge panels show the co-localization of AADC and TH. **b** Quantification of AADC+ (n = 5 WT, n = 6 *SMNΔ7*), TH+ (n = 7 WT, n = 6 *SMNΔ7*) and AADC+/TH+ cells in the *SNc* of *SMNΔ7* and WT mice. **c** Quantification of AADC+ (n = 6 WT, n = 6 *SMNΔ7*), TH+ (n = 7 WT, n = 6 *SMNΔ7)* and AADC+/TH+ cells in the VTA of *SMNΔ7* and WT mice. **d** Confocal images showing the labeling of DβH (red) and TH (green) in the *locus coeruleus (LC)* of *SMNΔ7* and WT mice at post-natal day 11 (n = 4/genotype). Nuclei were counterstained with Hoechst (blue). Merged panels show the co-localization of DβH and TH. **e** Quantification of DβH+, TH+, and DβH+/TH+ cells in the *LC*. **f** Confocal images showing the labeling of AADC (red) and TPH2 (green) in the *raphe nuclei* of *SMNΔ7* and WT mice at post-natal day 11 (n = 6/genotype). Nuclei were counterstained with Hoechst (blue). Merged panels show the co-localization of AADC and TPH2. **g** Quantification of AADC+, TPH2+ and AADC+/TPH2+cells. **a**, **d**, **f** Scale bars: **a** 200 μm, **d** 75 μm, **f** 10 μm. Dots represent individual mice values. *p < 0.05; **p < 0.01, ***p < 0.0001, compared to age-matched WT mice (unpaired *t*-test).

and T2, compared with T0 (Supplementary Fig. 8; Supplementary Table 7). Within each SMA type, we excluded specific associations with patients' age, BMI or motor achievements (Supplementary Table 8).

We would like to remark that this is a retrospective real-world study carried out on SMA patients recruited in 2018. Therefore, the clinical and biochemical information available during the longitudinal analysis was not always based on a strictly uniform sample size (Supplementary Tables 7–9), and no further parameters detailing the clinical state of the patients were available. For the same reasons, this study was not designed for DA determination, which is highly susceptible to oxidation when not protected by antioxidants at the time of collection[69,70].

Irrespectively of these limitations, our HPLC analysis shows a significant effect of SMN protein upregulation in inducing a significant and selective enhancement of NE levels in the CSF of severe SMA patients, thus highlighting a disease severity-specific neurochemical effect of Nusinersen treatment.

## Discussion

Multiorgan abnormalities observed in SMA patients and animal models are usually accompanied by severe glucose, fatty acids, and ketone body dysmetabolism[11–13,15–19,22,42,71]. However, it is not clear whether these metabolic dysfunctions depend on SMN deficiency or are secondary effects of diffuse denervation and what is their onset time.

Our untargeted NMR analysis in the *SMNΔ7* liver highlights that low SMN levels perturb metabolites involved in energy-related pathways, nucleotide, and amino acid metabolism since the neonatal stages of life. This data corroborate previous evidence in neonatal SMA mice showing a precocious influence of SMN deficiency on hepatic development and functioning[10,23,24]. Interestingly, at the late-symptomatic stage of the disease, the deregulated amino acid metabolism markedly converges on lower hepatic levels of tyrosine, phenylalanine, and isoleucine, likely reflecting abnormally enhanced metabolic consumption of ArAAs and BCAAs for fueling impaired glucose metabolism. Nonetheless, considering the overt feeding difficulties of SMA mice[50,72,73], we cannot exclude that inadequate nutritional intake of these essential amino acids may contribute to decreasing their hepatic levels at the latest stages of life. Additionally, in the liver of *SMNΔ7* mice at P11, untargeted

SMA1 patients (Supplementary Table 8). Moreover, we did not identify any significant 5-HT and 5-HIAA variation in SMA1 patients associated with Nusinersen-mediated improvement in motor outcomes (Supplementary Fig. 8; Supplementary Table 8). Like in SMA1 subjects, we found no Nusinersen-dependent changes in 5-HT and 5-HIAA concentrations in the CSF of milder SMA2 or SMA3 Nusinersen-treated patients, at both T1

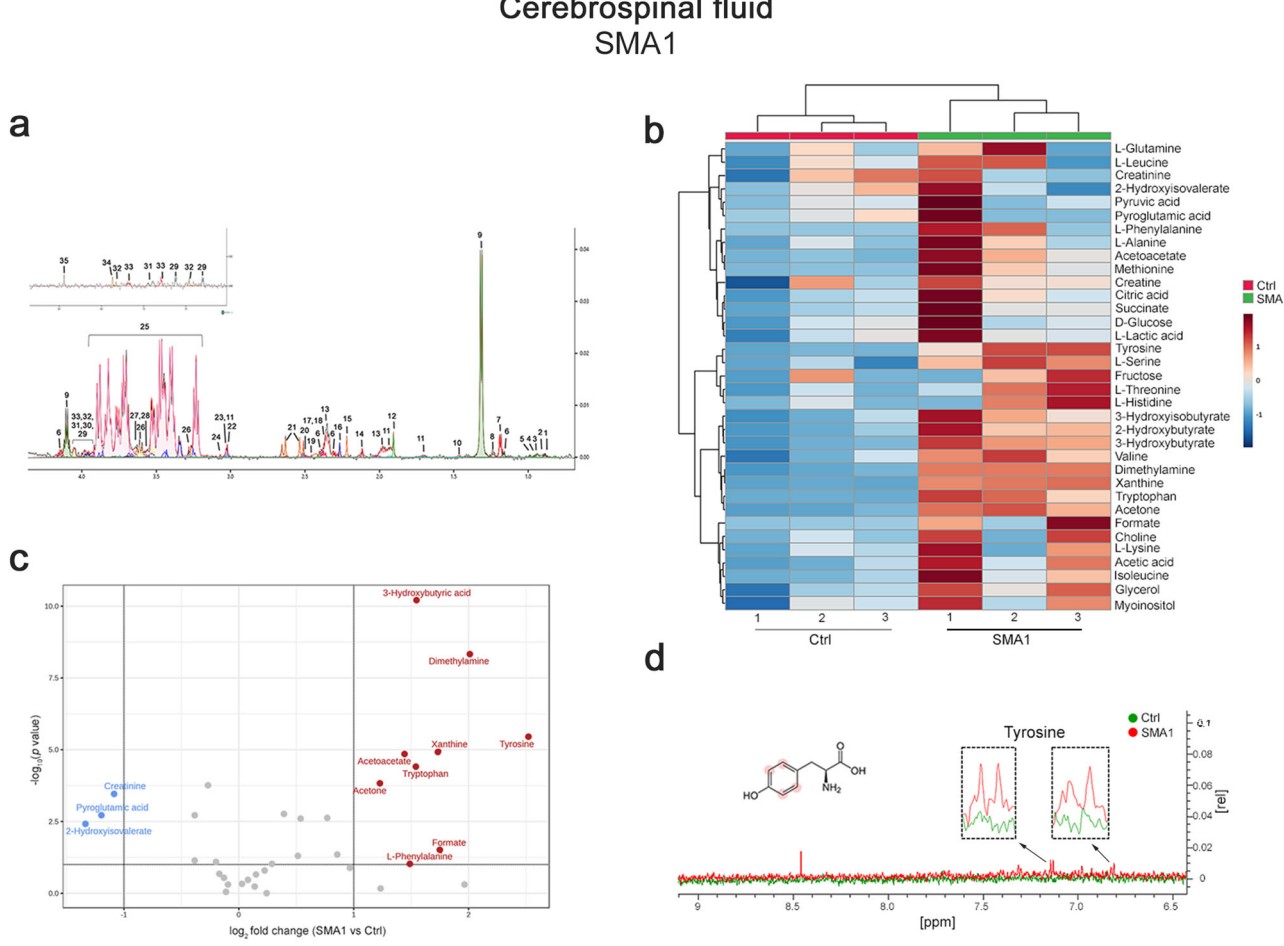

**Fig. 8 Untargeted NMR-based metabolome analysis in a cohort of naïve *SMA1* patients and healthy control subjects. a** Representative 1D $^1$H NOESY spectrum of SMA1 patient CSF extracts. The spectrum is acquired at 600 MHz and T = 310 K. Thirty-five CSF metabolites are identified and annotated as follow: 1: 2-Hydroxybutyric acid; 2: 2-Hydroxyisovalerate; 3: L-Leucine; 4: L-Isoleucine; 5: L-Valine; 6: 3-Hydroxybutyric acid; 7: 3-Hydroxyisobutyrate; 8: L-Threonine; 9: Lactic acid; 10: L-Alanine; 11: L-Lysine; 12: Acetic acid; 13: L-Glutamine; 14: L-Methionine; 15: Acetone; 16: Acetoacetate; 17: Pyroglutamic acid; 18: Succinic acid; 19: Pyruvic acid; 20: Dimethylamine; 21: Citric acid; 22: Creatine; 23: Creatinine; 24: Choline; 25: D-Glucose; 26: Myo-inositol; 27: Fructose; 28: Glycerol; 29: L-Tyrosine; 30: L-Serine; 31: L-Phenylalanine; 32: L-Histidine; 33: L-Tryptophan; 34: Xantine; 35: Formic acid. **b** Heatmap of changed metabolites. The color of each section corresponds to a concentration value of each metabolite calculated by a normalized concentration matrix (red, upregulated; blue, downregulated). **c** Volcano plot analysis of metabolic changes in naïve SMA1 patients (n = 3) and healthy control (Ctrl) (n = 3) CSF polar extracts. Each point on the volcano plot is based on both *p-value* and fold-change values, set at 0.05 and 2.0, respectively. Red points identify up-regulated metabolites whereas blue points identify down-regulated metabolites. **d** Superimposition of 1D $^1$H-NMR spectra of SMA1 patients (red) and Ctrl (green). The dashed boxes show the zooms of tyrosine signal in the superimposition. The proton signals displayed in the overlayed spectra are highlighted by red dots in the respective structures of the metabolite.

NMR data indicated reduced levels of malonate, which is the starting substrate for mitochondrial fatty acid biosynthesis, confirming the occurrence of lipid metabolism alteration in SMA[16]. Hence, our findings in *SMNΔ7* liver align with previous clinical and preclinical results indicating severe energy failure and mitochondrial dysfunctions in SMA[11–13,15–17,71].

Although previous studies describe that SMN deficiency significantly impacts brain development in SMA patients and animal models[54,55], the selective biochemical pathways affected by low cerebral SMN levels remain poorly characterized. Our untargeted NMR analysis disclosed that SMN deficiency dysregulates the brain metabolome of *SMNΔ7* mice, influencing aerobic glycolysis, pyruvate, and various amino acid metabolism, including BCAAs, ArAAs, glycine, and serine. However, unlike the liver, the cerebral metabolome abnormalities of *SMNΔ7* mice are restricted to the late-symptomatic phase. In agreement with these observations, preliminary NMR analysis in the CSF of naïve SMA1 patients and healthy subjects corroborates the influence of SMN

deficiency on the abnormal levels of phenylalanine, tyrosine, and other metabolites involved in energy metabolism, including ketone bodies.

Considering the pervasive alteration of ArAA metabolism observed in *SMNΔ7* mice, we assessed monoamine neurotransmitter levels in the CNS of these animals. Importantly, HPLC analysis showed a significant reduction of NE content in the brain and, to a lesser extent, the spinal cord of *SMNΔ7* mice. Coherently with neurochemical evidence, qRT-PCR experiments showed a prominent age-dependent *Dβh* mRNA downregulation in the brain of *SMNΔ7* mice, mirroring the lower DβH immunostaining seen in the adrenergic LC region of these animals. Although other studies are needed, we excluded that abnormal monoamine-regulating gene expression represents an epiphenomenon linked to motor neuron degeneration. Indeed, we found normal-like *Dβh and Aadc* transcript levels in the brain of symptomatic *SOD1* mice, a model of the degenerative neuromuscular disorder, ALS[74]. Noteworthy, we also found a not negligible reduction of noradrenergic neurons within the

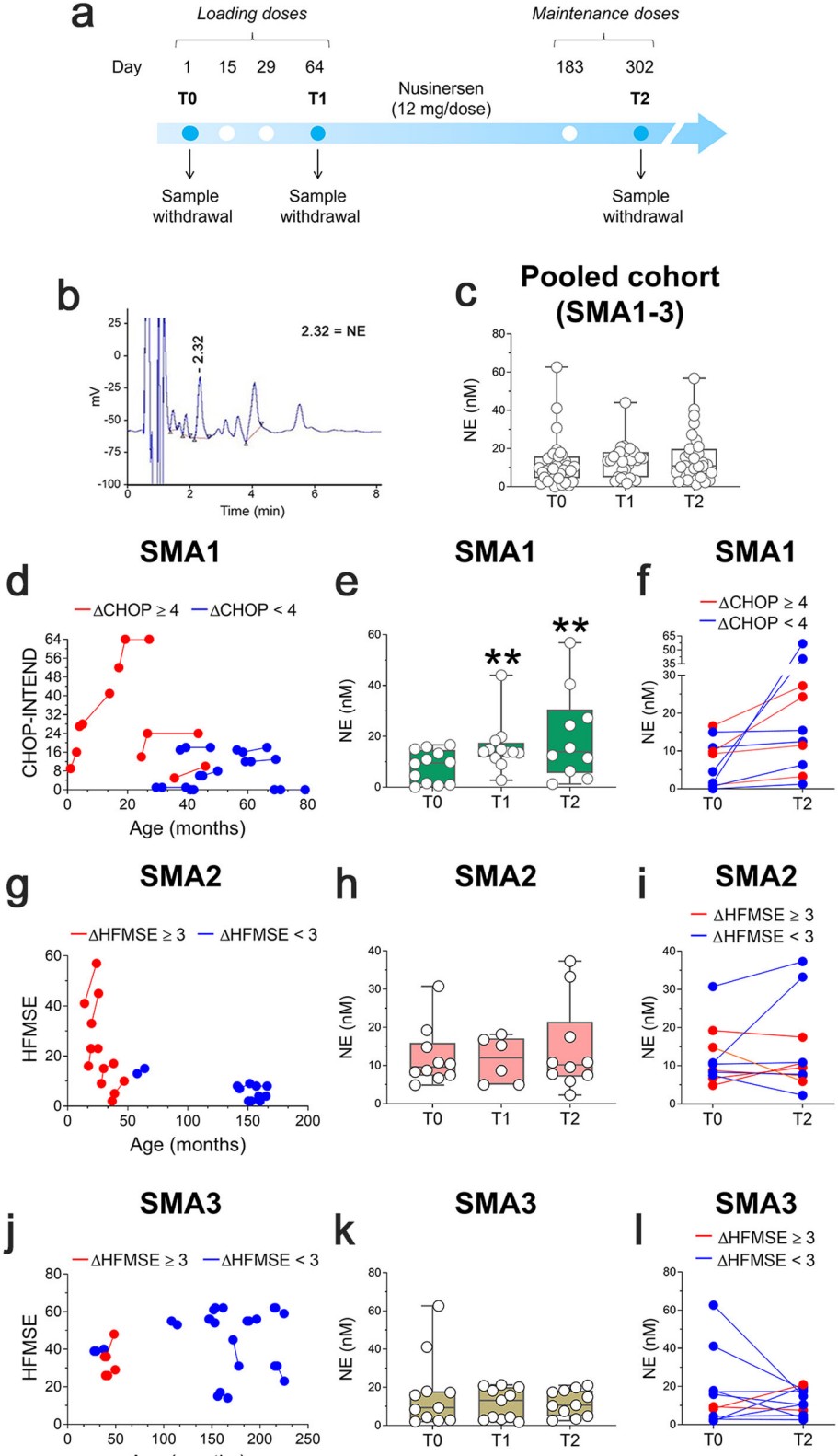

LC of *SMNΔ7* mice. Taken together, these results suggest that SMN deficiency may trigger cerebral NE neurotransmission impairments by affecting NE metabolism and survival of noradrenergic neurons in the *SMNΔ7* mouse brain. However, future studies in the *post-mortem* hindbrain specimens of severe patients must clarify this critical issue linking SMA pathology, neuroinflammation, and nor-adrenergic neuron degeneration in the LC.

As tyrosine is the main precursor of the biochemical chain leading to NE biosynthesis, we hypothesize that the DβH downregulation induced by SMN deficiency may hinder the utilization of this amino acid, leading to its abnormal cerebral accumulation in late-symptomatic *SMNΔ7* mice. Consistent with this scenario, we also found increased tyrosine levels in the CSF of naïve SMA1 patients compared to healthy subjects. Hence, we

**Fig. 9 Effect of Nusinersen on the CSF levels of norepinephrine in SMA1–3 patients. a** Schematic representation of the timeline of intrathecal Nusinersen administration and CSF collection in SMA patients. **b** Representative chromatogram of SMA1 patient showing norepinephrine (NE) peak and related retention time. **c** Levels of norepinephrine (NE) in the pooled cohort of SMA1, SMA2, and SMA3 patients (n = 33) prior to treatment (T0) and at the time of the fourth (T1, loading phase) and the sixth (T2, maintenance phase) injection of Nusinersen. **d** CHOP-INTEND scores of SMA1 patients (n = 12). **g**, **j** HFMSE scores of **g** SMA2 (n = 10) and **j** SMA3 (n = 11) patients. **e**, **h**, **k** Levels of NE in the CSF of **e** SMA1 (n = 12), **h** SMA2 (n = 10), and **k** SMA3 (n = 11) patients at T0, T1, and T2. **p < 0.01, compared to T0 (Wilcoxon matched-pairs signed ranks test). Data are shown as box and whisker plots representing the median with interquartile range (IQR). Dots represent individual patients' values. **f**, **i**, **l** *Spaghetti* plots representing variations of NE levels in the CSF of **f** SMA1, **i** SMA2, or **l** SMA3 patients between T0 and T2. Patients with ΔCHOP ≥ 4 or ΔHFMSE ≥ 3 are shown in red while patients with ΔCHOP < 4 or ΔHFMSE < 3 are shown in blue.

argue that the SMN induction brought by Nusinersen treatment may recover *DβH* gene expression favoring, in turn, the utilization of cerebral ArAAs, as indicated by their decreased levels in the CSF of treated SMA1 patients[42].

We found that SMN deficiency in the late-symptomatic *SMNΔ7* mouse brain triggers increased DA levels and TH phosphorylation at Ser40 residue, likely enhancing its enzymatic activity[75]. We interpret these changes as an adaptive mechanism aimed at counteracting the impaired DA to NE conversion caused by abnormally lower DβH levels in the *SMNΔ7* brain. Furthermore, our results show that late-symptomatic *SMNΔ7* mice display altered cerebral levels of AADC and MAO-A. Together with the regulation of DβH levels, these observations further support a modulatory role for SMN levels on the expression of the enzymes involved in monoamine metabolism[49], which fits well with the fundamental role of SMN in transcriptional and translational cell processes[1–3,6–8].

Viewed from a functional perspective, we revealed that the neurochemical abnormalities observed in the CNS of *SMNΔ7* mice appear mainly at an overt symptomatic stage of the disease when animals exhibit marked motor dysfunctions and weight loss[50,76] and are moribund due to their defective energy metabolism[15,18,22]. Therefore, considering the critical role of NE signaling in controlling neuronal excitability and energy metabolism through the stimulation of glycogenolysis in astrocytes[77–79], we propose that dysfunctional NE neurotransmission might represent a novel unrecognized primary factor contributing to the disease progression events associated with SMN deficiency. Coherently with the regulatory effect of SMN levels on NE production in the CNS of *SMNΔ7* mice, we reported a rapid and selective enhancement of this catecholamine in the CSF of SMA1 patients following Nusinersen-dependent SMN induction. In light of the implication of the NE system in the bioenergetic metabolism[77,79], we propose that the Nusinersen-dependent increase of central NE levels may contribute to the stimulation of glucose metabolism previously observed in the same cohort of SMA1 patients[42]. Within this frame of reasoning, our findings envisage a functional link between Nusinersen-mediated SMN induction, monoaminergic neurotransmission, and energy metabolism.

It is well-established that NE is a key regulator of neuroimmune responses, exerting both anti-inflammatory and neuroprotective effects through the activation of noradrenergic receptors located on peripheral immune cells, microglia, and astrocytes[62]. Accordingly, perturbation of NE signaling contributes to the pathophysiology of different inflammatory neurodegenerative disorders[62–64]. Based on these findings, we cannot exclude that the anti-inflammatory response elicited by Nusinersen treatment previously reported in the CSF of SMA1 patients[26] could rely on the upregulation of central NE levels identified in this study on the same patients' cohort.

Our metabolomic study in *SMNΔ7* mice unveiled that SMN deficiency induces a prominent ArAA and BCAA dysmetabolism, disclosing the risk of a deleterious hepatic depletion of these vital molecules in SMA, especially in the most severe forms, in which a

boost in amino acid consumption for fueling defective glucose metabolism is required[11,12,15,16,22,71]. In line with the present NMR data indicating a modulatory influence of SMN protein levels on ArAA and BCAA availability, we recently found that Nusinersen treatment strongly downregulates the CSF levels of these amino acids in SMA1 patients[42], probably due to their enhanced consumption in protein and monoamine biosynthesis, and energy metabolism. Therefore, these data envisage that exogenous supplementation of ArAAs and BCAAs might not only favor the hepatic metabolism of SMA patients but also potentiate the biochemical effects of Nusinersen in the CNS. In addition to their manifold roles[47,48,80], ArAAs, and BCAAs affect the physiological maturation of the CNS during the perinatal stages[81]. Therefore, correcting the peripheral metabolism of these essential amino acids may be crucial to avoid their harmful depletion triggered by SMN-inducing therapies and prevent CNS abnormalities in SMA patients, especially given their longer life expectancy.

The methodological power of NMR spectroscopy used in this study for metabolomic analysis provided both qualitative and quantitative untargeted identification of several metabolites in *SMNΔ7* mice and SMA patients. Nonetheless, in the perspective of an extension of our metabolomic studies, we propose to integrate the NMR-based approach with mass-spectroscopy analyses to further dissect metabolic misfunctioning associated with SMN deficiency[82,83].

Limitations of our retrospective study include: i) an overall limited number of SMA patients (n = 33), not supporting any additional speculation beyond those already advanced, especially regarding potential sex differences; ii) the lack of sex- and age-matched healthy controls for each SMA type. We aim to increase the number of SMA patients and matched control subjects in future prospective studies to verify the assumptions provided in the present study. Nevertheless, as SMA is a rare disease, there is an objective difficulty in assembling large SMA cohorts. Therefore, a more unified international approach to sample collection, processing, and data integration would be needed to provide significant advancement in this field. Finally, given the extent of the evolutionary and physiological differences between animal models and human diseases, we would like to advise that general caution is always necessary when interpreting dysfunctional metabolomic and neurochemical events occurring in patients in light of the findings obtained in animal models. Also due to this issue, future studies on *post-mortem* samples from SMA patients are advisable to confirm the hypotheses formulated in the present study.

In conclusion, our findings unveil a deleterious influence of SMN deficiency on liver and brain metabolism, prominently involving energy-related pathways and amino acid metabolism. In addition, to the best of our knowledge, we highlight for the first time a remarkable role for SMN deficiency in disrupting the expression of the main enzymes regulating central monoamine biosynthesis and degradation. These findings in SMA mice parallel the increase of NE levels in the CSF of SMA1 patients in response to Nusinersen-dependent SMN upregulation. Altogether, our results provide a milestone advance for understanding the regulatory role of SMN on

neurotransmission and the beneficial effects of SMN-inducing therapies at a neurochemical level. Moreover, they pave the way for novel therapeutic strategies targeting cerebral NE metabolism and nutritional approaches with selective amino acid supplementation as an add-on to the current SMA therapies.

## Materials and methods

**Mouse models and sample collection.** The original breeding SMN2[+/-]; SMNΔ7[+/-]; Smn[+/-] mice, heterozygous and healthy carrier for Smn gene mutation, were purchased from Jackson Laboratory (stock number 005025; Jackson Laboratories, Bar Harbor, ME, USA) and were bred to obtain Smn[+/+] (WT) and Smn[-/-] (SMNΔ7) animals. The offspring were genotyped by PCR assays on tail DNA according to the protocols as previously reported[84]. Data were obtained from male WT and SMNΔ7 mice sacrificed at P3, P6, and P11, considering P0 as the day of birth.

Regarding the ALS mouse model, the experiments were performed on male transgenic mice B6SJLTgN(SOD1G93A)1Gur overexpressing human SOD1, containing the Gly93Ala (G93A) mutation (The Jackson Laboratory, stock number 002726)[74]. The colony was derived by breeding male transgenic mice with naïve (B6xSJL/J)F1 females (WT) (The Jackson Laboratory, stock number 100012). Transgenic mice were identified by PCR as previously reported[85]. Data were obtained from WT and G93A mice sacrificed at four months.

Mice were housed in groups (n = 4–5) in standard cages (29 × 17.5 × 12.5 cm) at a constant temperature (22 ± 1 °C) and maintained on a 12 h light/dark cycle, with food and water ad libitum. Organs and tissues were collected from mice anesthetized with 1.5% sevoflurane, and 98.5% O$_2$ (Oxygen concentrator, Longfei Industry Co, Zhejiang, China). Brain, spinal cord, and liver were rapidly removed after cerebral dislocation and immediately frozen on dry ice and stored at −80 °C until use. Experiments were performed according to the international guidelines for animal research and approved by the Animal Care Committee of the University of Naples Federico II, Italy, and Ministry of Health, Italy. All efforts were made to minimize animal suffering and to reduce the number of animals. We have complied with all relevant ethical regulations for animal use.

**1H-NMR metabolomic analysis.** Whole brain and liver tissues were collected from WT and SMNΔ7 mice and processed according to the standard operating procedure[86]. In detail: 0.45 ml of water and 2 ml of methanol were added to 15–30 mg of each brain lyophilized tissue; moreover, 1 ml of water and 2 ml of methanol were added to 7–22 mg of liver lyophilized tissue. After sonication, the samples were diluted in 3 ml of a solution composed of chloroform/water in a 2/1 v/v ratio, then vortexed and centrifuged at 10,000 × g for 2 min at 4 °C. After centrifugation, a polar and an apolar phase were formed, which were separated and freeze-dried polar phase was analyzed via NMR. Dried polar extracts were dissolved into 500 μl of buffer (50 mM Na$_2$HPO$_4$, 1 mM trimethylsilyl propionic-2,2,3,3-d4 acid, sodium salt (TSP-d4), 50 μl of D$_2$O) and transferred into 5 mm NMR tubes for 1H-NMR detection.

CSF samples were analyzed as described by standard operating procedures[87]. Four hundred and twenty μl of buffer (50 mM of Na$_2$HPO$_4$, 1 mM of TSP-d4, and 50 μl of D$_2$O) was added to 80 μl of CSF and transferred to 5 mm NMR tubes for 1H-NMR detection. For both mouse and human studies, TSP-d4 0.1% in D$_2$O was used as an internal reference for aligning and quantifying NMR signals[88].

**NMR spectra acquisition and data analysis.** NMR experiments were acquired for all samples on a Bruker Ascend™ 600 MHz

spectrometer equipped with a 5 mm triple resonance Z gradient TXI probe (Bruker Co, Rheinstetten, Germany) at 298 K. Top-Spin, version 3.2 was used for the spectrometer control and data processing (Bruker Biospin). Liver, brain, and CSF Nuclear Overhauser Enhancement Spectroscopy (NOESY) 1D experiments were performed. Spectra acquisition was made using 14 ppm spectral width, 32k data points, presaturation during relaxation delay and mixing time for water suppression[89] and spoil gradient, 4 s relaxation delay, and mixing time of 10 ms. A weighted Fourier transform was applied to the time domain data with a line widening of 0.5 Hz followed by a manual step and baseline correction in preparation for targeted profiling analysis.

For CSF human samples, resonance assignment was carried out using Bayesil online server[90] by uploading the 1D 1H NOESY NMR spectra obtained from the CSF polar extracts of SMA1 patients. The software detected the presence of 35 metabolites and quantified them using as an internal standard the given concentration of TSP-d4[91]. The output matrix was used for statistical analysis.

Analysis of NMR spectra acquired on the brain and liver extracts of SMNΔ7 and WT mice was performed according to a targeted metabolomic approach. Specifically, qualitative research of metabolites was carried out using Chenomx NMR-Suite v8.0 (Chenomx Inc., Edmonton, Canada), which permits the identification of metabolites supported by the combination of advanced analysis tools with compound libraries. In particular, the analysis of spectra liver at P3 and P11 detected a number of 44 and 51 metabolites, respectively; moreover, the analysis of brain spectra identified 22 metabolites at both times. Subsequently, quantitative analysis was performed using NMRProcFlow[92], which allows multiple processing on 1D NMR spectra like baseline correction, ppm calibration, spectral alignment, solvent suppression, and normalization by probabilistic quotient. For the quantification, we chose ppm ranges for each metabolite peak located in uncrowded areas of the spectrum to avoid contamination by adjacent ones. All the data needed for the quantification were exported into a spreadsheet workbook using the "qHNMR" template, which collects information within five separate tabs: sample table, bucket table, signal-to-noise ratio matrix, and values of integration for each bucket (columns), and each spectrum (rows). The workbook includes sample volume (ml), the weight of brain or livers, the number of protons related to the metabolites' peaks, and the molecular weight of the metabolites, which are necessary for correct quantification. The data matrix generated by NMRProcFlow was finally used for statistical analysis.

**HPLC preparation of mouse tissue samples.** Frozen brain and spinal cord tissues were added with different volumes of 0.2 N perchloric acid (P11: 500 μl for the brain, 250 μl for the spinal cord; P6: 400 μl for the brain, 50 μl for the spinal cord; P3: 200 μl for the brain, 50 μl for the spinal cord). Tissues were homogenized with ultrasounds and subsequently centrifuged at 11,000 × g for 10 min at 4 °C. The supernatant was transferred into Spin-X microcentrifuge filters (0.22 μm nylon filter), centrifuged at 11,000 × g for 5 min at 4 °C and then stored at -80 °C until HLPC analysis.

**HPLC analysis.** Twenty μl samples (5 μl of hydrophilic phase from mouse samples or human CSF + 15 μl of 0.2 N PCA) were injected without purification into an HPLC equipped with a reverse phase column (LC-18 DB, 15 cm, 5 μm particle size, Supelco, Waters, Milford, MA, USA) and a coulometric detector (ESA, Coulochem II, Bedford, MA, USA) to quantify molecules such as L-DOPA, DA, NE, HVA, DOPAC, 5-HT, 5-HIAA in mouse samples, or NE, 5-HT and 5-HIAA in human CSF.

To detect catecholamines and their catabolites, the first electrode of the detector was set at +125 mV (oxidation) and the second at −175 mV (reduction). The composition of the mobile phase was (in mM): 50 NaH$_2$PO$_4$, 0.1 Na$_2$-EDTA, 0.5 n-octyl sodium sulfate, 15% (v/v) methanol, pH 3.7[93,94].

For 5-HT and 5-HIAA the first electrode of the detector was set at +280 mV (oxidation) and the second at −120 mV (reduction). The composition of the mobile phase was (in mM): 120 CH$_3$-COONa, 100 citric acid, 0.3 EDTA, 5% (v/v) methanol, pH 4.9[95]. For quantitative determination of each neurotransmitter, calibration curves were run using standards (SIGMA–ALDRICH, MO, USA). For data acquisition, the software ESA CDS (Euroservice, Ge, Italy) was used. Final values were expressed as nM (CSF) or fmol/mg of tissue (brain and spinal cord samples).

**Western blotting**. Total extracts were obtained in agreement with previous publication[96]. Tissues were lysed in a buffer containing 50 mM Tris-HCl, pH 7.4, 150 mM NaCl, 1 mM EDTA, 1% Triton X-100, and protease and phosphatase inhibitors. They were incubated for 1 h on ice and centrifuged at 10,000 rpm for 20 min. Supernatants were incubated with the following buffer: 62.5 mM Tris-HCl, pH 6.8, 2% sodium dodecyl sulfate (SDS), 10% glycerol, and 5% 2-mercaptoethanol. The extracts were electrophoresed through SDS-10% polyacrylamide gel. Only for P-TH-Ser40 and TH separation, SDS-12% polyacrylamide gel was used. The gels were electroblotted onto nitrocellulose membrane (Amersham Biosciences, Piscataway, NJ) in transfer buffer (48 mM Tris-HCl, 39 mM glycine, and 20% methanol). Specific primary antibodies were: anti-TH (mouse monoclonal antibody, 1:1000 mab-318; Millipore, Milan, Italy); anti-P-TH-Ser40 (rabbit polyclonal antibody, 1:1000; Cell Signaling 2791-S, Danvers, MA, USA); anti-AADC (rabbit polyclonal antibody, 1:1000; Novus Biological NB300-174, Centennial, CO, USA); anti-DβH (rabbit polyclonal antibody, 1:1000; Novus Biological NBP1-31386); anti-TPH2 (mouse monoclonal antibody, 1:1000; Abcam ab211528, Boston; MA, USA); anti-MAO-A (rabbit polyclonal antibody, 1:1000; Abcam ab126751); anti-MAO-B (rabbit polyclonal antibody, 1:1000; ab137778 Abcam); anti-COMT (rabbit polyclonal antibody, 1:1000; Abcam ab126618); anti-PAH (rabbit polyclonal antibody, 1:500; Novus Biological NBP-2-48615). Immunoreaction was revealed using antimouse and anti-rabbit immunoglobulin G conjugated to peroxidase 1:10,000 (GE Healthcare, Norwalk, CT) by the ECL reagent (GE Healthcare). The optical density of the bands was determined by ChemiDoc Imaging System (Biorad, Milan, Italy) and normalized to the optical density of β-Actin, used as an internal control.

**Quantitative real-time PCR**. Total RNA was extracted with Trizol following supplier's instructions (Life Technologies, Monza, Italy) and cDNA was synthesized using 2 μg of total RNA to obtain total cDNA with the High Capacity Transcription Kit following supplier's instruction (Life Technologies). Quantitative RT-PCR was performed with TaqMan assays in a 7500 qRT-PCR system (Life Technologies; ID: *Th* Mm00447557_m1; *Aadc* Mm00516688_m1; *Dβh* Mm00460472_m1; *Tph2* Mm00557715_m1; *Mao-a* Mm00558004_m1; *Mao-b* Mm00555412_m1; *Comt* Mm00514377_m1). Changes in mRNA levels were determined as the difference in threshold cycle ($2^{-\Delta\Delta Ct}$) between the target gene and an appropriate reference gene, *β-glucuronidase* (*Gusb* ID: Mm00446953_m1).

**Immunostaining and confocal immunofluorescence**. Mice were anesthetized and transcardially perfused[97]. Brains were removed and cryoprotected in 30% sucrose, frozen, and sectioned coronally at 40 μm on a cryostat in a rostral-caudal direction. Immunostaining and confocal immunofluorescence procedures were performed according to previous works[98,99]. After blocking with BSA 3%, brain sections were incubated overnight at 4 °C for 24 or 48 h with the following primary antibodies: anti-AADC (1:100, Abcam); anti-TH (1:250, Millipore); anti-TPH2 (1:250, Abcam). Sections were then incubated with the corresponding fluorescent-labeled secondary antibodies 1:300 (Alexa 488/Alexa 594 conjugated antirat/antimouse/antirabbit IgGs (Abcam and Jackson Immuno Research, Cambridge, UK). Nuclei were counterstained with Hoechst (Sigma-Aldrich, Milan, Italy). Images were observed using a Zeiss LSM700 META/laser scanning confocal microscope (Zeiss, Milan Italy). Digital images were taken with 10x and 60x objectives, an optical thickness of 0.7 μm, and a resolution of 1024 × 1024. Identical exposure times and laser power settings were applied to all the photographs from each experimental set.

**Cell-counting experiments**. Based on the anatomical levels of the *SNc* and VTA, images from the same areas of each brain region were compared, rostro-caudal sections were selected from the region 5.79 mm to 6.27 mm from rostral cortex[100] and included in the analysis. Single- and double-labeled cells (AADC$^+$, TH$^+$, AADC$^+$/TH$^+$) were counted on a total of seven matched rostral, medial, and caudal sections (same hemisphere) of the coronal midbrain, by using manual counting at 10 × magnification. Based on the anatomical levels of the LC, rostro-caudal sections were selected from the region 7.95 mm to 8.31 mm from rostral cortex[100]. Single- and double-labeled cells (DβH$^+$, TH$^+$, DβH$^+$/TH$^+$) were counted on a total of three matched sections (same hemisphere) of the coronal brain, using manual counting at 40 × magnification. Based on the anatomical levels of raphe *nuclei*, rostro-caudal sections were selected from the region 7.4 mm to 8.8 mm from the rostral cortex[100]. Single- and double-labeled cells (AADC$^+$, TPH2$^+$, AADC$^+$/TPH2$^+$) were counted on collected digital images (60 × magnification) within the raphe nuclei. Only cells with clearly visible cell body profiles were counted and included in the analysis. The average value of all the sections of each animal was determined. All immunostainings were blindly quantified.

**SMA patients' cohorts**. All subjects were enrolled at the Bambino Gesù Hospital (Rome, Italy). NMR metabolomic analysis was performed on three naive SMA1 patients and three pediatric controls (Supplementary Table 5). HPLC detections were performed on an independent cohort of 33 patients affected by SMA1 (n = 12), SMA2 (n = 10), and SMA3 (n = 11) who received intrathecal treatment with Nusinersen at the Bambino Gesù Hospital (Supplementary Table 6). The study was approved by the local Ethics Committee (2395_OPBG_2021). All ethical regulations relevant to human research participants were followed.

All participants and/or their legal guardians signed a written informed consent. All patients were clinically diagnosed and genetically confirmed, and *SMN2* copy number was also determined. All SMA1 patients, irrespective of their age and disease severity, were part of the Expanded Access Programme (EAP) for compassionate use to patients with the infantile form only, which occurred in Italy between November 2016 and November 2017. The overall clinical response of these patients to Nusinersen treatment has previously been reported as part of the full Italian cohort and showed that therapeutic efficacy is related to age and clinical severity at baseline[14,101]. The SMA2 and SMA3 patients have also been reported previously[102]. All 33 patients received injections of Nusinersen as *per* standard protocol. For this study, only CSF samples collected at day 0 (T0, baseline), day 64 (T1, at the time of the fourth Nusinersen

injection), and day 302 (T2, at the time of the sixth Nusinersen injection) were evaluated.

**Clinical characteristics of Nusinersen-treated SMA patients.** Assessment of patients was performed at baseline before the initiation of treatment (T0), at T1, and T2. At each visit, extensive clinical examination was performed by experienced child neurologists or pediatricians with expertise in the SMA field, anthropometric measurements, and vital parameters were collected. Moreover, patients' feeding status (oral nutrition, naso-gastric tube or percutaneous gastrostomy), nutritional status (postulated by BMI), and respiratory function (spontaneous breathing, NIV or tracheostomy) were recorded (Supplementary Table 8). All 12 SMA1 patients but two, were older than 1 year at the beginning of treatment with age ranging from 1 year and 7 months to 5 years and 7 months. Four patients had tracheostomy, 6 were under NIV for < 16 h/day and 2 patients were in spontaneous breathing. Ten patients had gastrostomy, and in all patients the BMI fell into the underweight range ( <18). The age of the 10 SMA2 patients ranged from 1 year and 2 months to 13 years and 3 months at baseline. Three of them were under NIV, none had gastrostomy, and the BMI was <18 in 5 patients. Regarding the 11 SMA3 patients, 3 were ambulant at baseline evaluation, one was under NIV for <16 h/day and none had gastrostomy. The BMI was <18 in 4 patients.

At T0, T1, and T2, all patients were assessed using standardized motor function tests, chosen according to their age and motor function. Functional assessments were performed by expert physiotherapists trained with standardized procedure manual[103] and reliability sessions. SMA1 patients were assessed with the CHOP-INTEND[66,67], a functional scale including 16 items that is aimed to assess motor function on weak infants. Each item is scored from 0 to 4 (with 0 being no response and 4 being complete level of response), with a total score ranging from 0 to 64. SMA2 and SMA3 patients were evaluated with the HFMSE[68], a scale of 33 items investigating the child's ability to perform different activities. The total score ranges from 0, if all the activities failed, to 66, indicating better motor function. All patients were not wearing spinal jackets or orthoses during the evaluations.

**Intrathecal treatment with Nusinersen.** Intrathecal administration of 12 mg of Nusinersen was performed in a hospital environment. A fasting <4 h was planned in advance of the procedure in SMA1 patients, while the time lapse between the last meal and the lumbar puncture was 6–8 h in SMA2 and SMA3 patients. The procedure was carried out without sedation whereas for SMA2 and SMA3 patients a sedation with midazolam was applied. No severe adverse events were reported. After the infusion, all patients were recommended to lie for 2 h to avoid any possible post-lumbar puncture symptoms.

**CSF sample collection of treated SMA patients.** CSF samples were collected at the time of the intrathecal administration of Nusinersen in polypropylene tubes and stored at −80 °C until further analysis. Determination of pH and total protein content was performed on the CSF sample of each patient (Supplementary Table 9). All CSF samples did not significantly differ for pH (SMA1, T0 *vs* T1, $p = 0.867$, T0 *vs* T2, $p = 0.826$; SMA2, T0 *vs* T1, $p > 0.999$, T0 *vs* T2, $p = 0.492$ ; SMA3, T0 *vs* T1, $p = 0.563$, T0 *vs* T2, $p = 0.378$; Wilcoxon test) and total protein content (SMA1, T0 *vs* T1, $p = 0.233$, T0 *vs* T2, $p = 0.275$; SMA2, T0 *vs* T1, $p > 0.999$, T0 *vs* T2, $p = 0.322$; SMA3, T0 *vs* T1, $p = 0.764$, T0 *vs* T2, $p = 0.206$; Wilcoxon test) across all time-points analyzed. Median values with minimum and maximum ranges of pH and total protein in patients' CSF are shown in Supplementary Table 9.

**Statistics and reproducibility.** Univariate analyses were performed using SPSS statistical software (IBM SPSS Statistics 20; Chicago, IL, USA) and GraphPad Prism 7.0 (GraphPad Software, La Jolla, CA, USA). HPLC and qRT-PCR experiments were performed in duplicates. Replicates used in cell-counting experiments are specified in the relevant paragraph. Detailed sample sizes are described in the figure legends. Normality distribution was assessed by the Kolmogorov–Smirnov and Shapiro-Wilk tests. Continuous variables were summarized as the median and interquartile range (IQR). Differences between mouse genotypes were analyzed by parametric unpaired t-test. The effect of mouse genotype and age, and their interaction was evaluated by two-way ANOVA. When the interaction was significant, simple main effects were reported and *post-hoc* tests were performed using the Benjamini-Hochberg (B-H) algorithm with False Discovery Rate (FDR) equal to 0.05[104,105]. Briefly, after sorting the p values of all *post-hoc* comparisons in ascending order and by assigning a rank i (from 1 to m, where m is the total number of comparisons), p values are considered statistically significant up to the largest *p-value* that is smaller than the critical value (i/m)xFDR. When the interaction was not significant, the main effects were instead evaluated and reported. Since sample sizes among Nusinersen groups are unequal, differences between >2 time-points (time after Nusinersen therapy) in CSF levels were evaluated by the Wilcoxon matched-pairs signed ranks test. Association between continuous variables was evaluated by Spearman's correlation, with correction for FDR as explained above.

NMR metabolomic data were analyzed by multivariate analysis (MVA). The One-Class Support Vector Machine (SVM) analysis has been conducted in order to identify the presence of outliers[106]. SVM data showed the absence of outliers (Supplementary Fig. 9). Matrices, including metabolites and their concentrations as derived from ${}^1$H-NMR data collected in 1D NOESY[89] were normalized by sum and using the Pareto scaling approach. The matrices were analyzed using a univariate approach combining t-test and fold-change through the Volcano plot. After normalization, data matrices were analyzed by principal component analysis (PCA), which reduces a dataset onto a lower-dimensional feature subspace while maintaining most of the relevant information, and PLS-DA, which reduces dimensionality while maximizing between-class separation, using MetaboAnalyst 5.0 (http://www.metaboanalyst. ca)[107–109]. The performance of the PCA and PLS-DA model was evaluated using the coefficient Q2 (using the 7-fold internal cross-validation method) and the coefficient R2, defining the variance predicted and explained by the model. The loading plots were used to identify the metabolites responsible for maximum separation in the PLS-DA score plot, and these metabolites were ranked according to their VIP scores, indicating the variables' importance[110]. To provide an intuitive visualization of data tables we performed heatmaps using normalized data, average groups concentration, and Euclidian distance[111]. The analysis of the pathways was carried out by analyzing the Enrichment tool. The output of Metaboanalyst was examined and the chosen KEGG paths according to the lower FDR, *p-value* < 0.05, and the hit values related to the number of metabolites belonging to the pathway>1.

**Reporting summary.** Further information on research design is available in the Nature Portfolio Reporting Summary linked to this article.

## Data availability

NMR data have been deposited to the EMBL-EBI MetaboLights database (DOI: 10.1093/ nar/gkz1019, PMID:31691833) with the identifier MTBLS8784. Raw HPLC, Western blotting, qRT-PCR, and immunohistochemistry data are provided in the form of 3 Excel data files in Supplementary Material (Supplementary Data 1–3). Uncropped immunoblot

images from Fig. 5 and Supplementary Fig. 3–5 are available as Supplementary Fig. 10–13 in the pdf file "Supplementary information". All data needed to evaluate the conclusions in the paper are present in the paper and in the Supplementary Materials.

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

## Acknowledgements

We thank Anders Borgkvist, Carmelina Melone, Emanuela Santini, Marcello Serra, David Sulzer, Matteo Vidali, and Alessandro Zazza for their useful discussion and help.

A.U. G.P. T.N., E.B., and A.D.A. were supported by #NEXTGENERATIONEU (NGEU) funded by the Ministry of University and Research (MUR), National Recovery and Resilience Plan (NRRP), project MNESYS (PE0000006) – A Multiscale integrated approach to the study of the nervous system in health and disease (DN. 1553 11.10.2022). E.B. and A.D.A. were also supported by a grant from Ricerca Finalizzata from the Italian Ministry of Health (Project nr RF-2019-12370334); E.B. and A.D.A. are members of the ERN NMD European Network (Project nr 2016/557).

## Author contributions

V.V. and G.L. worked with *SMNΔ7* and *SOD1* mouse colony, performed western blotting and qRT-PCR experiments, and acquired data. P.B. and A.C. performed IHC experiments and acquired data. V.V. and G.P. supervised western blotting, qRT-PCR and IHC analyses, and data acquisition. V.B. performed HPLC experiments and acquired data. M.C. supervised HPLC analysis and data acquisition. C.M. and M.G. performed NMR experiments and acquired data; A.M.D.U. supervised NMR analysis and data acquisition. T.N. performed biochemical analyses in the CSF, analyzed data, and prepared the figures. A.D.A. and E.B. performed clinical evaluations and provided CSF samples. A.U. and F.E. wrote and edited the paper. A.U. conceived and designed the study.

## Competing interests

The authors declare no competing interests.
