## [Peer Review File · Communications Biology]

Reviewers' comments:

Reviewer #1 (Remarks to the Author):

The authors Usiello et al. submitted a manuscript titled "SMN deficiency perturbs catecholamine metabolism in spinal muscular atrophy".

The manuscript is well structured, contains novel approaches, detailed chemical analyses and sufficient statistical interrogation of the underlying data.

The overall structure of the manuscript containing interesting novel information is acceptable. Statistical methods are of standard quality, but need some additional descriptions.

The following comments would need to be addressed before the manuscript could be deemed acceptable for publication:

1. Statistical methods

Please describe in more detail why different approaches for data evaluation were utilized while using data from different origins, as this information provides significant framework for readers to understand why different approaches were utilized. I understand that usually there are different groups performing different types of analyses and then data are compiled and integrated. However, the basic reasons for different data analyses such as data structure, types of variables, multivariate vs univariate, distributions need to be presented, so readers can understand that.

2. Size of the sample utilized in this study

Usually the results obtained from such small samples result in significant overfitting and overexplanation of the information present in the data. Please provide a general comment at the end of the manuscript that would put the results presented in your study in a general context, taking into account that things might be well much different when the size of the data is larger for an order of magnitude and uniformity of the underlying cohorts are increased, and contrasted with matched cohorts, as these things are usually missing from the manuscripts in general.

3. Materials and methods

After reading MM section several times, I feel that these section would need to be amended by additional information, that would enable other researchers to use and comply with the same analytical approaches as utilized in this study. That would definitely improve the stringency of the research in this field. In the end, the field of SMA is hampered by the fact that many groups utilized contrasting analytical approaches that in actual experience obscure the underlying information and themselves skew the results in certain direction. With this in mind, a general statement towards more unified analytical approaches should be placed within this manuscript.

For instance, the exact version of False Discovery Rate (FDR) method of Benjamini and Hochberg is not reported, therefore a citation would be useful.

4. Human vs animal models

A general statement of caution on the extent of the differences between animal models of human diseases needs to become part of the revised manuscript in order to sufficiently convey the information that while the animal models are useful, the underlying evolutionary differences and physiological differences do not substitute them for the actual ethically acceptable research on human materials.

5. Spell check

Please run your ms through the spell check program as some minor typos still exist.

Please revise sentences that are of excessive length, and hence to understand in uniform manner.

6. Gender Equality Problems

Please provide a general statement that male and female differences in SMA have not been sufficiently addressed so far, and that your current sample size does not support any additional speculation, before this is resolved with cohorts utilizing larger samples size.

7. Table S6

Supplementary Table 6. Clinical characteristics of SMA1 patients enrolled in the study at fourth (T1. loading phase) and sixth (T2. maintenance phase) injection of Nusinersen.

The manuscript should provide a cautionary statement that the underlying patient cohort is actually not strictly uniform, and that other explanatory parameters could contribute to this end.

Also, the sample size dropped from 12 to 10, which also needs a cautionary statement on the dropout rate of the overall experiment.

Also, the NMR healthy controls were 3 in comparison to 12 SMA, that would deserve another comment of caution not to overextrapolate the comments.

8.

As SMA is a rare disease, undoubtedly there is significant difficulty to assemble large SMA cohorts, therefore a more EU wide unified approach on sample collection, processing and data integration would be needed to provide significant advancement in this field. This paper could serve as a starting venue to address this issue in the future with a larger collaborative approaches, to integrate over various genetic, geographic and other characteristics.

9.

Biochemical mechanisms proposed in this study on SMA1 patients not need to be present and replicated in other SMA types, therefore caution would need to be exercised while interpreting these results and manuscript should well address this as well.

Overall, the manuscript contains valuable information, however it needs to be rephrased as some sections so as not to overstate the actual value, as clinicians usually focus on the stated results and skip the warning usually contained within the manuscript MM sections. So these pieces of information need to be provided within the manuscript clearly.

Reviewer #2 (Remarks to the Author):

This is a review for the manuscript by Professor Usiello and colleagues on "SMN deficiency perturbs catecholamine metabolism in spinal muscular atrophy". The authors performed a multi-omics study in SMN Δ 7 mice (liver and brain) and CSF of naïve SMA1 patients, and they have identified by NMR-based metabolomics some metabolic irregularities related to energy homeostasis pathways and aromatic amino acid metabolism, along with specific decrease of mRNA and protein levels of aromatic L-amino acid decarboxylase, dopamine beta-hydroxylase and monoamine oxidase A. I considered this a complete study, and the conclusions are well-supported by a vast range of experimental procedures allowing to detect disturbances in mRNA, proteins and metabolites. Please see specific some questions and concerns to be addressed in the numbered list below.

1. About the Introduction: I think the introduction is too short and important information that is crucial for the understanding of the article is missing. I did not find an explanation on the aims and purpose of the study: what was done, either in animal models as in patients, and why, and what was the methodology employed. The authors only refer in the last chapter: "Our investigations indicate that SMN deficiency modulates monoamine metabolism in both the severe SMA mouse model, SMN Δ 7 17,18 97, and the CSF of SMA1 patients after Nusinersen therapy." This sentence is more a resume of the results obtained than an explanation of the aim of the article. For example, there is no mention of the analysis of the liver and brain of SMN Δ 7 mice. The authors refer in the abstract that "findings in animal models and patients also indicate multiorgan and metabolic abnormalities." But why were the liver and the brain chosen and analyzed by NMR untargeted metabolomics approach? Why not other organs?

2. About the Results: Explain abbreviation for BCAA (Branched Chain Amino Acids) the first time you wrote – I think it was in line 115.

3. Figures 1g, 2g and 7g. place the disturbed metabolites above the corresponding peaks at the spectra or arrows indicating their position in the spectra.

4. Figure 2. The downfield region of the spectrum is missing (in Figure 1a the authors showed both regions). If it is not necessary (I see that assignation of arAAs was also made in the upfield region), please specify in the legend the spectral region (in ppm) shown of the NMR spectrum.

5. Line 281. "3-hydroxybutyric acid" replace by "3-hydroxybutyric acid".

6. The last model for naïve SMA1 patients is indeed with a very low n (n=3 for each group). I think with this n, information retrieved by this model may not be reliable (despite the authors refer R²=0.67 and Q²=0.89 and that the model was validated by cross-validation...). In these cases of low n, univariate data analysis is preferred. The authors conducted the volcano plot, so the results seem valid. Nevertheless, I would not add the VIP scores plot from this model.

7. About the Methods: I would prefer to read the methods by order of appearance of the results. In the results section, the results on animal models were explained first, and then the ones with patients. The authors refer "Afterwards, to translate the preclinical results collected in SMN Δ 7 mice to a clinical setting, we extended the untargeted NMR-based metabolomic analysis to the CSF of naïve SMA1 patients and age-matched healthy subjects." So, for an easier understanding and reading of the article, I would maintain this same order in all the sections of the methods.

8. Line 584. "Multivariate data analysis (MVA) was carried out by MetaboAnalyst 5.0 to identify the metabolomic profile of the brain and liver polar solvent extract of SMN Δ 7 and WT mice at P11." Please reformulate the sentence: Metaboanalyst does not allow to identify the metabolomics profile, but disturbances on the metabolomic profile after application of MVA methods.

9. Line 595: "The loading plot" – replace by "loadings plot".

10. Concerning PCA and PLS-DA models, what was done in case outliers were detected? Also, which scaling method was employed for each model?

11. Line 563. For CSF human data, which kind of normalization was applied?

12. Line 597. "The output of Metaboanalyst was examined and the KEGG paths were chosen that presented the rate of false discoveries (FDR) lower, p-value 0.05, and the hits value related to the number of metabolites belonging to the pathway > 1." Please reformulate the sentence: "...and the chosen KEGG paths presented the ..."

13. Section "Statistical analysis." Please specify the software(s) employed.

14. On the Discussion. Line 354. Please mention the pitfalls of the study and the low n.

15. Line 409. The authors refer that "independent golden standard analytical methods (i.e., mass spectrometry) will be desirable in future studies to reliably quantify the variation of specific metabolites in both human and animal model samples of SMA disease". Please explain why NMR is not a suitable method to quantify the biomarkers found in this study (ArAAs and BCAAs, among others). Is this meaning that the quantifications obtained in the study are not reliable?

Reviewer #1

The authors Usiello et al. submitted a manuscript titled "SMN deficiency perturbs catecholamine metabolism in spinal muscular atrophy". The manuscript is well structured, contains novel approaches, detailed chemical analyses and sufficient statistical interrogation of the underlying data. The overall structure of the manuscript containing interesting novel information is acceptable. Statistical methods are of standard quality, but need some additional descriptions.

We are pleased the Reviewer finds that this a well-structured manuscript containing detailed analyses and novel approaches and information in the SMA field. We have addressed the specific points raised by the Reviewer as described below:

1. Statistical methods

Please describe in more detail why different approaches for data evaluation were utilized while using data from different origins, as this information provides significant framework for readers to understand why different approaches were utilized. I understand that usually there are different groups performing different types of analyses and then data are compiled and integrated. However, the basic reasons for different data analyses such as data structure, types of variables, multivariate vs univariate, distributions need to be presented, so readers can understand that.

R: We thank the Reviewer for her/his valuable suggestion. Accordingly, the statistical analysis section was completely rewritten with the aim to improve both understanding and reproducibility: 1) the section was divided into a univariate and a multivariate analysis; 2) all tests applied were reported in the section; 3) when skew or far from gaussian distribution was present, non-parametric tests were used: indeed comparing genotypes with parametric t-test was possible but when comparing time-points non-parametric tests were needed (the specific situation was reported together with the test used); 4) a detailed section on the two-way ANOVA test was added; 5) the correction for *post-hoc* comparison was briefly described and two references were added; 6) the type of SVM used to detect outliers was added; 7) the aim of both PCA and PLS-DA was added. We hope that all these modifications may help add homogeneity to the section, increasing the reader's understanding.

2. Size of the sample utilized in this study

Usually the results obtained from such small samples result in significant overfitting and overexplanation of the information present in the data. Please provide a general comment at the end of the manuscript that would put the results presented in your study in a general context, taking into account that things might be well much different when the size of the data is larger for an order of magnitude and uniformity of the underlying cohorts are increased, and contrasted with matched cohorts, as these things are usually missing from the manuscripts in general.

R: Based on the critical Reviewer's observation, in the present version of the work we have included also CSF samples from 10 SMA2, and 11 SMA3 patients, bringing the total number of SMA patients from 12 to 33. This allowed us to evaluate the neurochemical effects of Nusinersen on a higher number size of patients (SMA1-3) and, moreover, to discriminate the influence of this medication according to the disease severity. Notwithstanding, we agree with the Reviewer that the limited number of patients enrolled in our study cannot support definitive clinical conclusions, which is to be expected in a real-world retrospective analysis performed on a rare disease such as SMA. Therefore, taking advantage of

the Reviewer's suggestion, we have now remarked that this issue is a limitation of our work and highlighted the necessity to perform further studies on higher sample sizes that include also gender- and age-matched cohorts.

(Text: "Limitations of our retrospective study include: i) an overall limited number of SMA patients (n=33), not supporting any additional speculation beyond those already advanced, especially regarding potential gender differences; ii) the lack of gender- and age-matched healthy controls for each SMA type. Regarding the small sample size here used, we aim to increase the number of SMA patients and matched control subjects in future prospective studies to verify the assumptions provided in the present study.").

3. Materials and methods

After reading MM section several times, I feel that these sections would need to be amended by additional information, that would enable other researchers to use and comply with the same analytical approaches as utilized in this study. That would definitely improve the stringency of the research in this field. In the end, the field of SMA is hampered by the fact that many groups utilized contrasting analytical approaches that in actual experience obscure the underlying information and themselves skew the results in certain direction. With this in mind, a general statement towards more unified analytical approaches should be placed within this manuscript.

For instance, the exact version of False Discovery Rate (FDR) method of Benjamini and Hochberg is not reported, therefore a citation would be useful.

R: We agree with the Reviewer about the importance to specify in great detail the methodological description of the experiments performed in a paper with the aim to improve the reproducibility of data. In the new version of the manuscript, the statistical section was completely rewritten; more details about all tests used were added. Tests were reported together with their scenario (e.g. unpaired t-test for genotype groups, Wilcoxon test when comparing time-points, two-way ANOVA for age x genotype). A brief description of the method of Benjamini and Hochberg, together with 2 references, was added. All modifications were made taking into account the reproducibility of results.

4. Human vs animal models

A general statement of caution on the extent of the differences between animal models of human diseases needs to become part of the revised manuscript in order to sufficiently convey the information that while the animal models are useful, the underlying evolutionary differences and physiological differences do not substitute them for the actual ethically acceptable research on human materials.

R: We agree with the Reviewer that findings in animal models should be generally taken with caution when trying to predict events occurring in human diseases. In this regard, we have now specified in the Discussion section that the hypotheses on the metabolomic and neurochemical mechanisms in SMA, formulated in light of the findings obtained in mouse models, need further confirmation in patients ("Finally, given the extent of the evolutionary and physiological differences between animal models and human diseases, we would like to advise that general caution is always necessary when interpreting dysfunctional metabolomic and neurochemical events occurring in patients in light of the findings obtained in animal models. Also due to this issue, future studies on *post-mortem* samples from SMA patients are advisable to confirm the hypotheses formulated in the present study.").

5. Spell check

Please run your ms through the spell check program as some minor typos still exist.

Please revise sentences that are of excessive length, and hence to understand in uniform manner.

R: We thank the Reviewer for her/his suggestion. We have now used a spell check program that allowed us to hopefully eliminate typing errors and enhance the comprehension of the text.

6. Gender Equality Problems

Please provide a general statement that male and female differences in SMA have not been sufficiently addressed so far, and that your current sample size does not support any additional speculation, before this is resolved with cohorts utilizing larger samples size.

R: We are aware that assessing gender differences in SMA would have been an intriguing point of our study. However, as acknowledged by the Reviewer, the low sample size of our retrospective work did not allow us to address this topic. Thanks to the Reviewer's remark, we have now pointed out this issue as a limitation of the present work (see the text reported at point 2 of the rebuttal: "2. *Size of the sample utilized in this study*").

7. Table S6

Supplementary Table 6. Clinical characteristics of SMA1 patients enrolled in the study at fourth (T1. loading phase) and sixth (T2. maintenance phase) injection of Nusinersen.

The manuscript should provide a cautionary statement that the underlying patient cohort is actually not strictly uniform, and that other explanatory parameters could contribute to this end.

Also, the sample size dropped from 12 to 10, which also needs a cautionary statement on the dropout rate of the overall experiment.

Also, the NMR healthy controls were 3 in comparison to 12 SMA, that would deserve another comment of caution not to overextrapolate the comments.

R: We thank the Reviewer for her/his accurate reading of the Tables and the Results sections. We would like to highlight that the non-uniformity of the patients' sample size at different therapy stages and the lack of further "explanatory parameters" is a quite common limitation in real-world retrospective studies on severe diseases such as SMA. However, we took advantage of the Reviewer's comment to provide a cautionary statement that clinical and biochemical parameters of patients are not strictly uniform ("We would like to remark that this is a retrospective real-world study carried out on SMA patients recruited in 2018. Therefore, the clinical and biochemical information available during the longitudinal analysis was not always based on a strictly uniform sample size (Suppl. Tables 5-7), and no further parameters detailing the clinical state of the patients were available.").

Moreover, we would like to remark that in the present study, the NMR analysis on humans was performed on CSF samples from an equal number of healthy subjects and untreated SMA1 patients (n=3 subjects/clinical condition). On the other hand, we recruited an independent cohort of SMA1 patients (n=12), before and after Nusinersen treatment, for HPLC detections. We have now better specified this point in the new Methods section ("NMR metabolomic analysis was performed on three naive SMA1 patients and three pediatric controls (Table 1). HPLC detections were performed on an independent cohort of 33 patients affected by SMA1 (n=12), SMA2 (n=10) and SMA3 (n=11) who received intrathecal treatment with Nusinersen at the Bambino Gesù Hospital (Table 2)."). However, we agree with the Reviewer that multivariate analyses performed for the NMR analysis are not a reliable analytical

model when dealing with such a limited number of subjects (n=3). Therefore, in the present manuscript version, we removed the relevant PLS-DA, VIP score and Pathway enrichment analysis.

8. As SMA is a rare disease, undoubtedly there is significant difficulty to assemble large SMA cohorts, therefore a more EU wide unified approach on sample collection, processing and data integration would be needed to provide significant advancement in this field. This paper could serve as a starting venue to address this issue in the future with a larger collaborative approaches, to integrate over various genetic, geographic and other characteristics.

R: We find it interesting that the Reviewer considers that our study may serve as a venue to propose new worldwide multi-center collaborations for clinical studies regarding such a rare disorder as SMA. Taking advantage of this cue, in the new version of the Discussion we launched the idea that “a more unified international approach to sample collection, processing and data integration would be needed to provide a significant advancement in this field.”.

9. Biochemical mechanisms proposed in this study on SMA1 patients not need to be present and replicated in other SMA types, therefore caution would need to be exercised while interpreting these results and manuscript should well address this as well.

R: The Reviewer correctly raised the point that our neurochemical analyses in the CSF of Nusinersen-treated SMA1 patients missed a comparison with other SMA types. Therefore, the biochemical mechanisms we proposed to explain the cerebral effects of Nusinersen on monoamine neurotransmitters did not consider the potential disease severity-dependent effects of this medication. Based on this pertinent observation, in the new version of the manuscript, we performed further HPLC analyses of monoamine levels also in the CSF samples of SMA2 and SMA3 patients, which were still available in our lab. Thanks to this Reviewer’s criticism, we have now demonstrated that Nusinersen exerts neurochemical effects on monoamine neurotransmitters depending on the disease severity of SMA patients.

Overall, the manuscript contains valuable information, however it needs to be rephrased as some sections so as not to overstate the actual value, as clinicians usually focus on the stated results and skip the warning usually contained within the manuscript MM sections. So these pieces of information need to be provided within the manuscript clearly.

R: We thank the Reviewer for her/his overall suggestions. Now we have rephrased several unclear sentences and described in more detail the methodological sections, trying not to overstate the actual value of the manuscript. Moreover, we believe that now we have greatly improved the quality of the manuscript thanks to the Reviewer’s suggestion to extend our research to the analysis of other disease severity conditions, namely SMA2 and SMA3 types.

Reviewer #2

This is a review for the manuscript by Professor Usiello and colleagues on “SMN deficiency perturbs catecholamine metabolism in spinal muscular atrophy”. The authors performed a multi-omics study in SMNΔ7 mice (liver and brain) and CSF of naïve SMA1 patients, and they have identified by NMR-based metabolomics some metabolic irregularities related to energy homeostasis pathways and aromatic amino acid metabolism, along with specific decrease of mRNA and protein levels of aromatic L-amino acid decarboxylase, dopamine beta-hydroxylase and monoamine oxidase A. I considered this a complete study, and the conclusions are well-supported by a vast range of experimental procedures allowing to detect disturbances in mRNA, proteins and metabolites. Please see specific some questions and concerns to be addressed in the numbered list below.

We greatly appreciate the Reviewer’s overall comments on the quality and relevance of our study for the SMA field. We are delighted that she/he considered our study complete and the conclusions well supported by a wide range of experimental procedures. We have addressed her/his specific comments as described below:

1.About the Introduction: I think the introduction is too short and important information that is crucial for the understanding of the article is missing. I did not find an explanation on the aims and purpose of the study: what was done, either in animal models as in patients, and why, and what was the methodology employed. The authors only refer in the last chapter: “Our investigations indicate that SMN deficiency modulates monoamine metabolism in both the severe SMA mouse model, SMNΔ7 17,18 97, and the CSF of SMA1 patients after Nusinersen therapy.” This sentence is more a resume of the results obtained than an explanation of the aim of the article. For example, there is no mention of the analysis of the liver and brain of SMNΔ7 mice. The authors refer in the abstract that “findings in animal models and patients also indicate multiorgan and metabolic abnormalities.” But why were the liver and the brain chosen and analyzed by NMR untargeted metabolomics approach? Why not other organs?

R: We thank the Reviewer to have noticed that the Introduction was too short and lacked several background information that is important for a better comprehension of the manuscript’s purposes and results. Accordingly, in the present version of the manuscript, we have significantly extended the Introduction section describing further topics relevant to the context of this manuscript, including i) the complex and manyfold physiological roles of the SMN protein, ii) the recent wider picture of SMA as a complex multi-organ and metabolically relevant disorder (beyond motor neuron implication), iii) the molecular mechanism of Nusinersen action, iv) the limitation of the current SMA therapies, including the lack of comprehensive studies assessing their possible metabolic and neurochemical effects.

Thanks to the Reviewer’s inquiry, we believe that these changes have greatly enhanced the structure of the Introduction providing a wider overview of the state of the art of this complex disorder and a clearer comprehension of the aims and purposes of our work. Moreover, as suggested by the Reviewer, we have removed the last sentence of the Introduction, which sounded like a kind of repetitive summary of the work.

2.About the Results: Explain abbreviation for BCAA (Branched Chain Amino Acids) the first time you wrote – I think it was in line 115.

R: We have now used this acronym at the first mention.

3. Figures 1g, 2g and 7g. place the disturbed metabolites above the corresponding peaks at the spectra or arrows indicating their position in the spectra.

4. Figure 2. The downfield region of the spectrum is missing (in Figure 1a the authors showed both regions). If it is not necessary (I see that assignment of arAAs was also made in the upfield region), please specify in the legend the spectral region (in ppm) shown of the NMR spectrum.

R: We thank the Reviewer for noticing such inaccuracies in the Figures. In the new version of these Figures, we have placed arrows indicating the position of the boxed peaks and included the downfield regions of the NMR spectra.

5. Line 281. “3-hydroxybutyric acid” replace by “3-hydroxybutyric acid”.

9. Line 595: “The loading plot” – replace by “loadings plot”.

R: Now we removed these typos.

6. The last model for naïve SMA1 patients is indeed with a very low n ($n=3$ for each group). I think with this n , information retrieved by this model may not be reliable (despite the authors refer $R^2=0.67$ and $Q^2=0.89$ and that the model was validated by cross-validation...). In these cases of low n , univariate data analysis is preferred. The authors conducted the volcano plot, so the results seem valid. Nevertheless, I would not add the VIP scores plot from this model.

R: As correctly noticed by the Reviewer, we realized that multivariate analyses are not a reliable analytical model when dealing with such a limited number of subjects. Therefore, in the present manuscript version, we removed the PLS-DA, VIP score and Pathway enrichment analysis.

7. About the Methods: I would prefer to read the methods by order of appearance of the results. In the results section, the results on animal models were explained first, and then the ones with patients. The authors refer “Afterwards, to translate the preclinical results collected in SMN Δ 7 mice to a clinical setting, we extended the untargeted NMR-based metabolomic analysis to the CSF of naïve SMA1 patients and age-matched healthy subjects.” So, for an easier understanding and reading of the article, I would maintain this same order in all the sections of the methods.

R: We agree with the Reviewer that the description sequence of the methods should follow the order of appearance of the results. We are sorry for the improper presentation of this section. We have now ordered the Methods section according to the description of the results.

8. Line 584. “Multivariate data analysis (MVA) was carried out by MetaboAnalyst 5.0 to identify the metabolomic profile of the brain and liver polar solvent extract of SMN Δ 7 and WT mice at P11.” Please

reformulate the sentence: Metaboanalyst does not allow to identify the metabolomics profile, but disturbances on the metabolomic profile after application of MVA methods.

R: As requested by the Reviewer, we changed the sentence to improve the clarity.

10. Concerning PCA and PLS-DA models, what was done in case outliers were detected? Also, which scaling method was employed for each model?

R: We thank the Reviewer for raising such an important question. We have now specified that, before carrying out the multivariate analysis, a classification method with Support Vector Machine (SVM) was carried out using a confusion matrix to assess the presence of outliers. As mentioned in the new version of the Methods section, no outliers emerged from the class prediction probability analysis and the confusion matrix. For greater clarity, we have also inserted a new Supplementary Figure (Suppl. Fig. 9) showing the SVM results.

11. Line 563. For CSF human data, which kind of normalization was applied?

R: CSF data were normalized by sum, log, and Pareto scaling. We apologize to the Reviewer for omitting this important information that, in the present version of the manuscript, has been included in the Methods section.

12. Line 597. "The output of Metaboanalyst was examined and the KEGG paths were chosen that presented the rate of false discoveries (FDR) lower, p-value 0.05, and the hits value related to the number of metabolites belonging to the pathway > 1." Please reformulate the sentence: ... "and the chosen KEGG paths presented the ..."

R: We reformulated the sentence as requested by the Reviewer.

13. Section "Statistical analysis." Please specify the software(s) employed.

R: We have now specified the software used for our statistical analyses.

14. On the Discussion. Line 354. Please mention the pitfalls of the study and the low n.

R: In agreement with the Reviewer's request, we have now included a section describing the limitation of our work. We are confident that this change will greatly enhance the criticism consistency of our Discussion ("Limitations of our retrospective study include: i) an overall limited number of SMA patients (n=33), not supporting any additional speculation beyond those already advanced, especially regarding potential gender differences; ii) the lack of gender- and age-matched controls for each SMA type. Regarding the limited sample size here used, we aim to increase the number of SMA patients and

matched control subjects in future prospective studies to verify the assumptions provided in the present study. Nevertheless, as SMA is a rare disease, there is an objective difficulty to assemble large SMA cohorts. Therefore, a more unified international approach to sample collection, processing and data integration would be needed to provide significant advancement in this field. Finally, given the extent of the evolutionary and physiological differences between animal models and human diseases, we would like to advise that general caution is always necessary when interpreting dysfunctional metabolomic and neurochemical events occurring in patients in light of the findings obtained in animal models. Also due to this issue, future studies on *post-mortem* samples from SMA patients are advisable to confirm the hypotheses formulated in the present study.”).

15. Line 409. The authors refer that “independent golden standard analytical methods (i.e., mass spectrometry) will be desirable in future studies to reliably quantify the variation of specific metabolites in both human and animal model samples of SMA disease”. Please explain why NMR is not a suitable method to quantify the biomarkers found in this study (ArAAs and BCAAs, among others). Is this meaning that the quantifications obtained in the study are not reliable?

R: We are grateful to the Reviewer for raising this question because we realized that we have given the wrong meaning to this sentence. We have now reformulated the phrase as follows: “The methodological power of NMR spectroscopy used in this study for metabolomic analysis provided qualitative and quantitative untargeted identification of several metabolites in SMNΔ7 mice and SMA patients. Nonetheless, in the perspective of an extension of our metabolomic studies, we propose to integrate the NMR-based approach with mass-spectroscopy analyses to further dissect metabolism malfunctioning associated with SMN deficiency^{82,83}.”

REVIEWERS' COMMENTS:

Reviewer #2 (Remarks to the Author):

The authors correctly addressed my questions and performed all the modifications asked. I have no other concerns and I believe this article is suitable for publication